# Enhancing calmodulin binding to cardiac ryanodine receptor completely inhibits pressure-overload induced hypertrophic signaling

Michiaki Kohno[1,4], Shigeki Kobayashi[1,4], Takeshi Yamamoto[2,4], Ryosuke Yoshitomi[1], Toshiro Kajii[1], Shohei Fujii[1], Yoshihide Nakamura[1], Takayoshi Kato [1], Hitoshi Uchinoumi[1], Tetsuro Oda[1], Shinichi Okuda [1], Kenji Watanabe[3], Yoichi Mizukami[3] & Masafumi Yano[1✉]

Cardiac hypertrophy is a well-known major risk factor for poor prognosis in patients with cardiovascular diseases. Dysregulation of intracellular $Ca^{2+}$ is involved in the pathogenesis of cardiac hypertrophy. However, the precise mechanism underlying cardiac hypertrophy remains elusive. Here, we investigate whether pressure-overload induced hypertrophy can be induced by destabilization of cardiac ryanodine receptor (RyR2) through calmodulin (CaM) dissociation and subsequent $Ca^{2+}$ leakage, and whether it can be genetically rescued by enhancing the binding affinity of CaM to RyR2. In the very initial phase of pressure-overload induced cardiac hypertrophy, when cardiac contractile function is preserved, reactive oxygen species (ROS)-mediated RyR2 destabilization already occurs in association with relaxation dysfunction. Further, stabilizing RyR2 by enhancing the binding affinity of CaM to RyR2 completely inhibits hypertrophic signaling and improves survival. Our study uncovers a critical missing link between RyR2 destabilization and cardiac hypertrophy.

[1] Department of Medicine and Clinical Science, Division of Cardiology, Yamaguchi University Graduate School of Medicine, 1-1-1 Minamikogushi, Ube, Yamaguchi 755-8505, Japan. [2] Faculty of Health Sciences, Yamaguchi University Graduate School of Medicine, 1-1-1 Minamikogushi, Ube 755-8505, Japan. [3] Institute of Gene Research, Yamaguchi University Science Research Center, Yamaguchi 755-8505, Japan. [4]These authors contributed equally: Michiaki Kohno, Shigeki Kobayashi, Takeshi Yamamoto. ✉email: yanoma@yamaguchi-u.ac.jp

Cardiac hypertrophy is a well-known major risk factor for poor prognosis in patients with cardiovascular diseases. Pressure-overload initially causes hypertrophy, which is usually regarded as a compensatory response to normalize wall stress, whereas prolonged hypertrophy is regarded as detrimental, promoting increased oxygen consumption and cardiomyocyte death[1]. In contrast, several experimental studies have indicated that abolishment of the hypertrophic response to systolic pressure overload does not result in left ventricle (LV) decompensation, but rather in beneficial outcomes such as improved contractile function and prognosis[2,3]. These findings obviously question the compensatory nature of initial hypertrophy in response to pressure overload.

Alterations in $Ca^{2+}$ handling, such as $Ca^{2+}$ leakage through RyR2 influence hypertrophic signaling and electrical remodeling[4,5], and pharmacological inhibition of $Ca^{2+}$ leakage led to attenuation of cardiac hypertrophy[5]. Based on the domain-switch hypothesis proposed by Ikemoto and his colleagues[6], we have shown that defective inter-domain interactions between the N-terminal domain (a.a. 1–600) and the central domain (a.a. 2000–2500) of RyR2 (domain unzipping), causes $Ca^{2+}$ leakage through RyR2, inducing catecholaminergic polymorphic ventricular tachycardia (CPVT) and heart failure[7–10]. We also showed that domain unzipping causes a leaky $Ca^{2+}$ channel via dissociation of CaM from RyR2[11,12], and that RyR2 function was restored by the introduction of modified CaM (+ Gly-Ser-His), which has a significantly higher binding affinity for RyR2[12]. These findings strongly suggest that in the pathogenesis of heart failure, domain unzipping is allosterically coupled with conformational changes in the CaM binding domain (3583–3603), leading to CaM dissociation and subsequent $Ca^{2+}$ leakage. The essential role of CaM on the pathogenesis of heart failure was clearly shown by the finding that amino acid substitutions within a core CaM binding sequence in RyR2 (W3587A/L3591D/F3603A)} that disrupts CaM binding, indeed resulted in severe cardiac hypertrophy and early death in mutant mice[13]. Although the zipping-unzipping hypothesis can largely explain the pathogenic mechanism of $Ca^{2+}$ leakage in CPVT or heart failure, high-resolution structures determined by cryo-electron microscopy (cryo-EM) revealed that the CPVT-causing mutations do not target a single interface, instead of affecting multiple small domain-domain interfaces, within and across the hotspots[14,15]. In particular, most disease mutations in the N-terminal region of RyR1 and 2 involve inter-domain interfaces within the N-terminal region, and only a small fraction may interact with the central hot spots[16]. Therefore, with the atomic level structures in mind, more rigorous analysis is needed to determine the critical role of inter-domain interactions on RyR2 function in heart failure.

Recently, we generated a heterozygous $RyR2^{V3599K/+}$ mouse model in which a single point mutation in the CaM binding domain of RyR2 was inserted, to increase the binding affinity of CaM for RyR2. These mice were crossed with $RyR2^{R2474S/+}$ mice to generate double heterozygous $RyR2^{R2474S/V3599K}$ mice to examine whether an enhanced binding affinity of CaM to RyR2 may rescue from CPVT[17]. Heterozygous $RyR2^{R2474S/V3599K}$ mice showed no CPVT, in contrast to $RyR2^{R2474S/+}$ mice, clearly indicating that genetic enhancement of CaM binding affinity to RyR2 by a single amino-acid substitution rescues CPVT-associated arrhythmogenesis, characterized by bidirectional or polymorphic ventricular tachycardia, spontaneous $Ca^{2+}$ transients, and $Ca^{2+}$ sparks[17].

Based on these findings, we used homozygous $RyR2^{V3599K/V3599K}$ (V3599K) mice to examine the role of CaM-RyR2 complex in the development of pressure-overload induced cardiac hypertrophy.

Our results showed that genetic rescue of CaM dissociation from RyR2 prevents $Ca^{2+}$ leakage, thereby protecting against cardiac hypertrophy and improving prognosis.

## Results

**RyR2 V3599K mutation suppressed the development of cardiac hypertrophy.** At baseline conditions before transverse aortic constriction (TAC), there was no appreciable difference in the structural or functional features of the heart between wild-type (WT) and V3599K mice (Supplementary Fig. 1a, Fig. 1a–d). Two weeks after TAC, both LV wall thickness and LV weight increased in WT mice but not in V3599K mice, and 8 weeks after TAC, the LV chamber was markedly enlarged showing hypertrophy and reduced contractile function (Fig. 1a, b, d). After TAC, interstitial fibrosis was observed in WT cardiomyocytes but not in V3599K cardiomyocytes (Fig. 1a). Such structural and functional changes were significantly reduced in V3599K mice (Fig. 1a, b, d). Peak systolic pressure increased to a similar extent after TAC in both WT and V3599K mice, indicating that a comparable degree of pressure-overload was imposed on LV (Supplementary Fig. 1b, Fig. 1c). In WT mice, although the dP/dt max of LV pressure was maintained 2 weeks after TAC, both dP/dt min and time constant (Tau) of LV pressure decay were already worse (Supplementary Fig. 1b, Fig. 1c). Of particular interest, survival was markedly higher in V3599K mice than in WT mice (Fig. 1e). To assess how LV adapts to chronic pressure-overload, a LV pressure-volume relationship was obtained during a rapid fall in systolic pressure upon occlusion of inferior vena cava. The slope of LV end-systolic pressure-volume relationship (Ees) decreased 8 weeks after TAC in WT mice, indicating decreased LV contractility. In contrast, Ees became rather steeper 8 weeks after TAC in V3599K mice than in Sham (V3599K) mice, indicating increased LV contractility (Fig. 2).

**RyR2 V3599K mutation improved intracellular $Ca^{2+}$ kinetics.** Consistent with these in vivo data, hypertrophy of cardiomyocytes was observed after TAC in WT cardiomyocytes but not in V3599K cardiomyocytes (Fig. 3a, b). No significant differences were observed in the contour or kinetic parameters of $Ca^{2+}$ transient or sarcomere shortening at baseline between WT and V3599K cardiomyocytes (Fig. 3c, d). Two weeks after TAC, sarcomere shortening and peak $Ca^{2+}$ transients were comparable between WT and V3599K mice. However, the time from peak to 70% decline in both sarcomere shortening and $Ca^{2+}$ transient was already prolonged in WT cardiomyocytes, relative to V3599K cardiomyocytes (Fig. 3c, d). Eight weeks after TAC, sarcomere shortening was reduced, the peak of $Ca^{2+}$ transient decreased, and duration was prolonged in WT cardiomyocytes, whereas all these parameters were ameliorated in V3599K cardiomyocytes (Fig. 3c, d).

In WT cardiomyocytes, the $Ca^{2+}$ spark frequency increased even 2 weeks after TAC, which may cause prolonged $Ca^{2+}$ transient, whereas these abnormalities were restored in V3599K cardiomyocytes (Fig. 4a, b). The sarcoplasmic reticulum (SR) $Ca^{2+}$ content decreased 8 weeks after TAC in WT cardiomyocytes, whereas it did not decrease in V3599K cardiomyocytes (Fig. 4c).

Taken together, these results suggest that the initial, so-called "compensatory" hypertrophy against pressure-overload appears to be a maladaptive response, characterized by abnormal $Ca^{2+}$ cycling toward heart failure. The results also suggest that the prevention of hypertrophy from the beginning maintains cardiac function under healthy conditions, although wall stress would be increased based on the Laplace law.

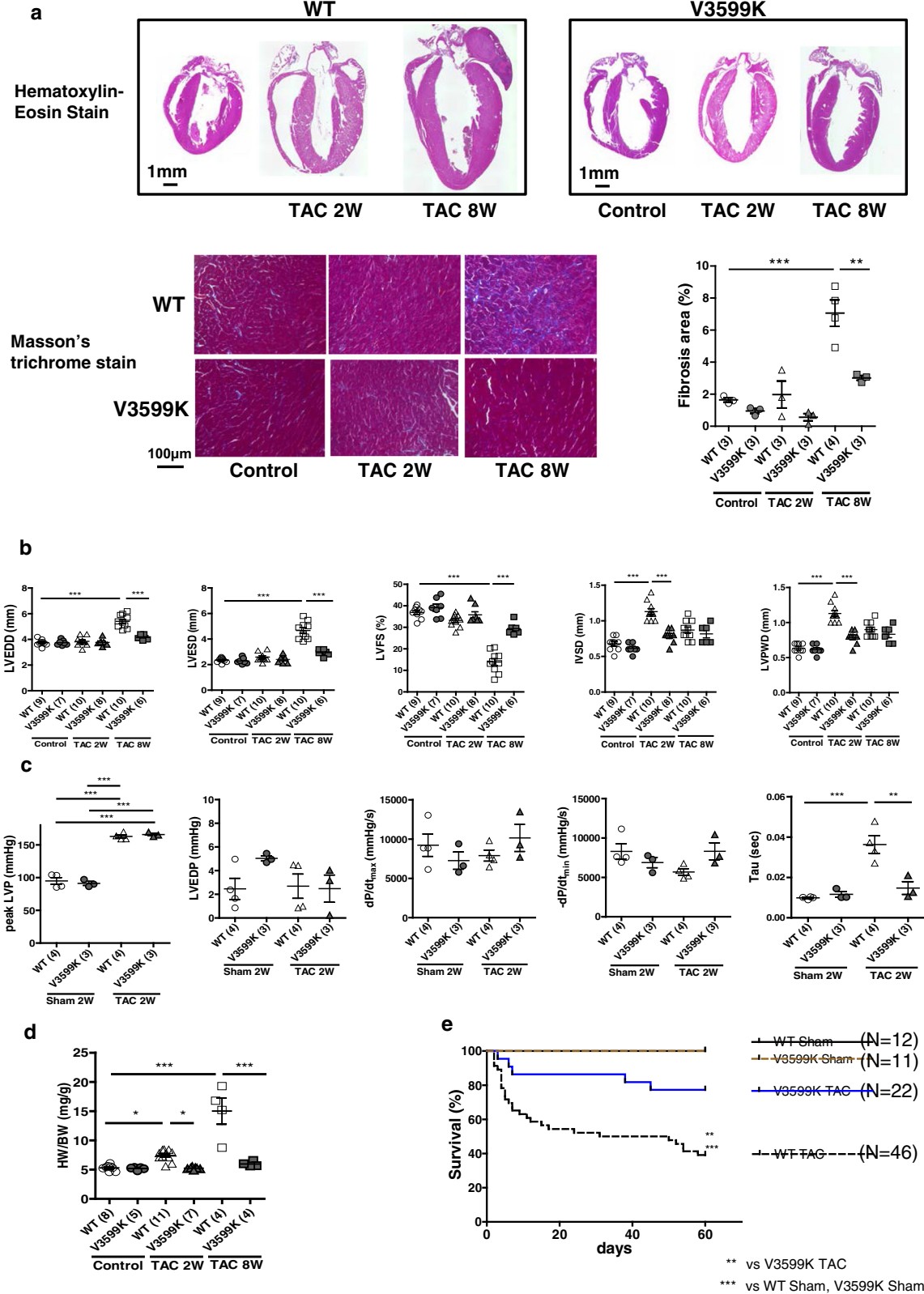

**Fig. 1 Structural and functional characteristics, and prognosis after TAC in WT and V3599K mice. a** Representative images of long axis sections of the hearts, and hematoxylin and eosin or Masson's trichrome stained LV tissue. **b** LV end-diastolic diameter (LVEDD), LV end-systolic diameter (LVESD), LV fractional shortening ((LVEDD − LVESD)/LVEDD × 100), intra-ventricular septum diastolic thickness (IVSD), left ventricular posterior wall diastolic thickness (LVPWD). **c** Peak LVP, LVEDP, dP/dt max of LVP, dP/dt min of LVP, Tau. **d** LV weight/body weight. **e** A Kaplan–Meier survival analysis. Values for individual mice are plotted together with mean ± SEM. Parentheses indicate the number of mice. *$P < 0.05$, **$P < 0.01$, ***$P < 0.001$ (one-way ANOVA with post-hoc Tukey's multiple comparison test).

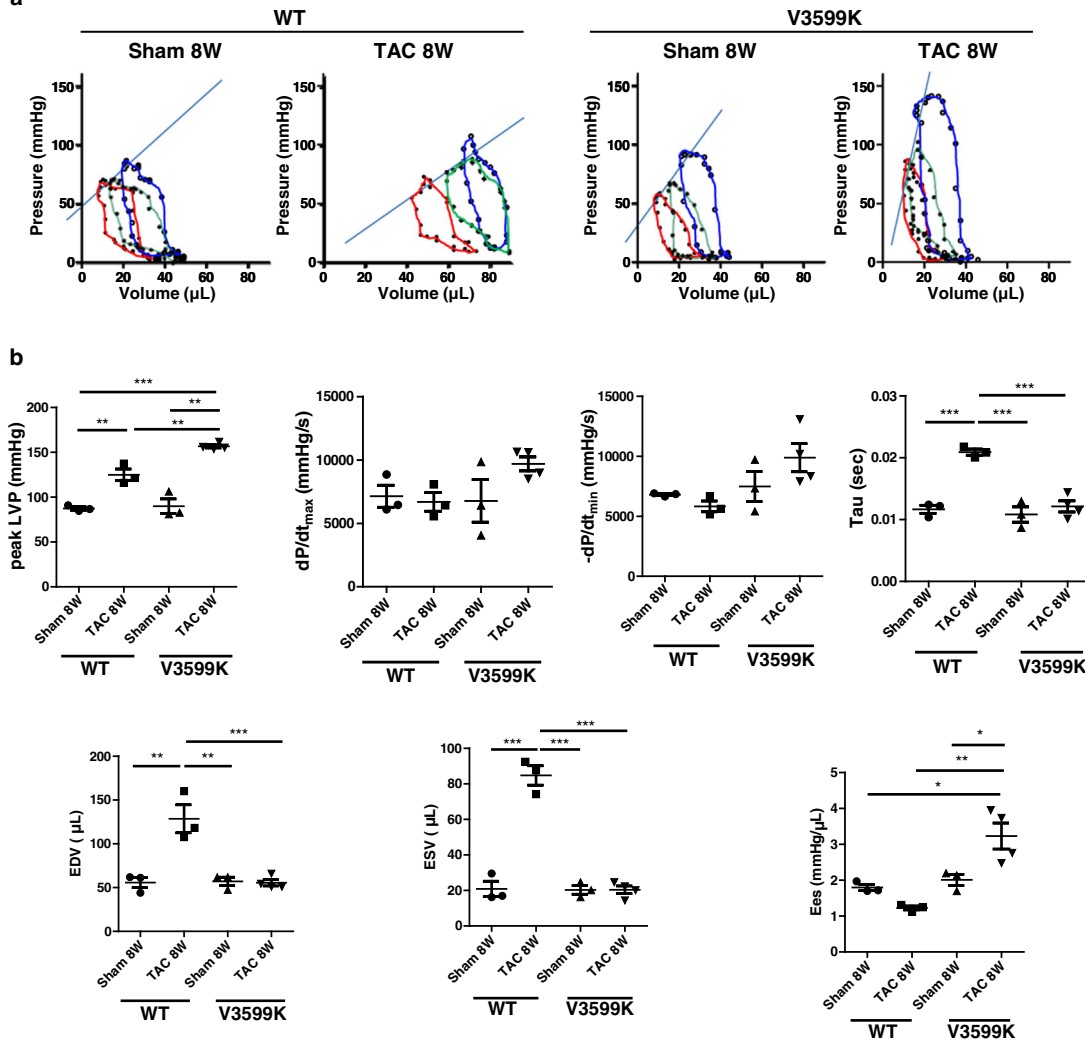

**Fig. 2 LV pressure (P) –volume (V) relationship after chronic pressure-overload. a** Representative P-V loops. **b** Hemodynamic parameters: Peak LVP, dP/dt max of LVP, dP/dt min of LVP, Tau. LV end-diastolic volume (LVEDV), LV end-systolic volume (LVESV), and Ees. *$P < 0.05$, **$P < 0.01$, ***$P < 0.001$ (one-way ANOVA with post-hoc Tukey's multiple comparison test).

**Chronic pressure-overload displaced CaM from RyR2 in WT, but not in V3599K mice.** To further elucidate the role of intracellular $Ca^{2+}$ handling on the characteristics of the hypertrophic myocardium, we evaluated the association of CaM to RyR2 and $Ca^{2+}$ release kinetics in cardiomyocytes. Endogenous CaM was co-localized with RyR2 on the Z-line in WT cardiomyocytes before TAC (Supplementary Fig. 2a, Fig. 4d). In contrast, it was displaced after TAC in WT cardiomyocytes. In V3599K cardiomyocytes endogenous CaM was normally associated with RyR2 along the Z-line (Fig. 4d). There was no difference in the endogenous CaM between control (before TAC). Sham 2 W, and Sham 8w, in WT and V3599K mice (Supplementary Fig. 2b). Direct binding of exogenous CaM to RyR2 was also evaluated by attaching a UV cross-linker to CaM. The binding affinity of CaM to RyR2 was markedly decreased after TAC in WT hearts, whereas it was restored in V3599K hearts (Fig. 4e). Suramin (10 μM), which directly binds to the core RyR-CaM binding sequence and displaces CaM from both RyR1 and RyR2 channel isoforms[18,19], indeed inhibited CaM binding to RyR2, confirming the specificity of CaM binding to RyR2.

**Enhanced CaM binding to RyR2 reduced the expression of hypertrophy-related genes.** Whole transcriptome analysis using RNA-seq analysis in the hearts from WT and V3599K mice with or without TAC was performed to elucidate the RyR2 signaling pathway in the heart. In the heart, more than 19,000 genes were detected from approximately 27 million reads in each sample. Among RyR superfamily, RYR2 was mainly expressed in mice hearts, and the expressions were almost constant in the treated mice (Supplementary Fig. 3). The expression of *Calm1*, which encodes CaM was enhanced with TAC in WT mice hearts, and expression was disappeared from those in V3599K mice (Supplemental Fig. 3). Principal component analysis (PCA) was performed to cluster the hearts according to gene expression patterns. In the PCA plot, V3599K {without (−TAC) or with TAC (+TAC)} and WT mice {without (−TAC) or with TAC (+TAC)} were clearly separated by PC2. In PC3, that WT (−TAC) and WT (+TAC) show distinct gene expression patterns, whereas V3599K (−TAC) and V3599K (+TAC) were not separated (Fig. 5a). These results indicate that the gene expression pattern is similar between V3599K (−TAC) and V3599K (+TAC), whereas WT (−TAC) and WT (+TAC) differ significantly. The heatmap shows the gene expression of the upper 100 genes indicated by factor loadings of PC3. Gene expression increased in WT (+TAC) hearts compared to WT (−TAC) hearts; however, this increase was not observed in V3599K

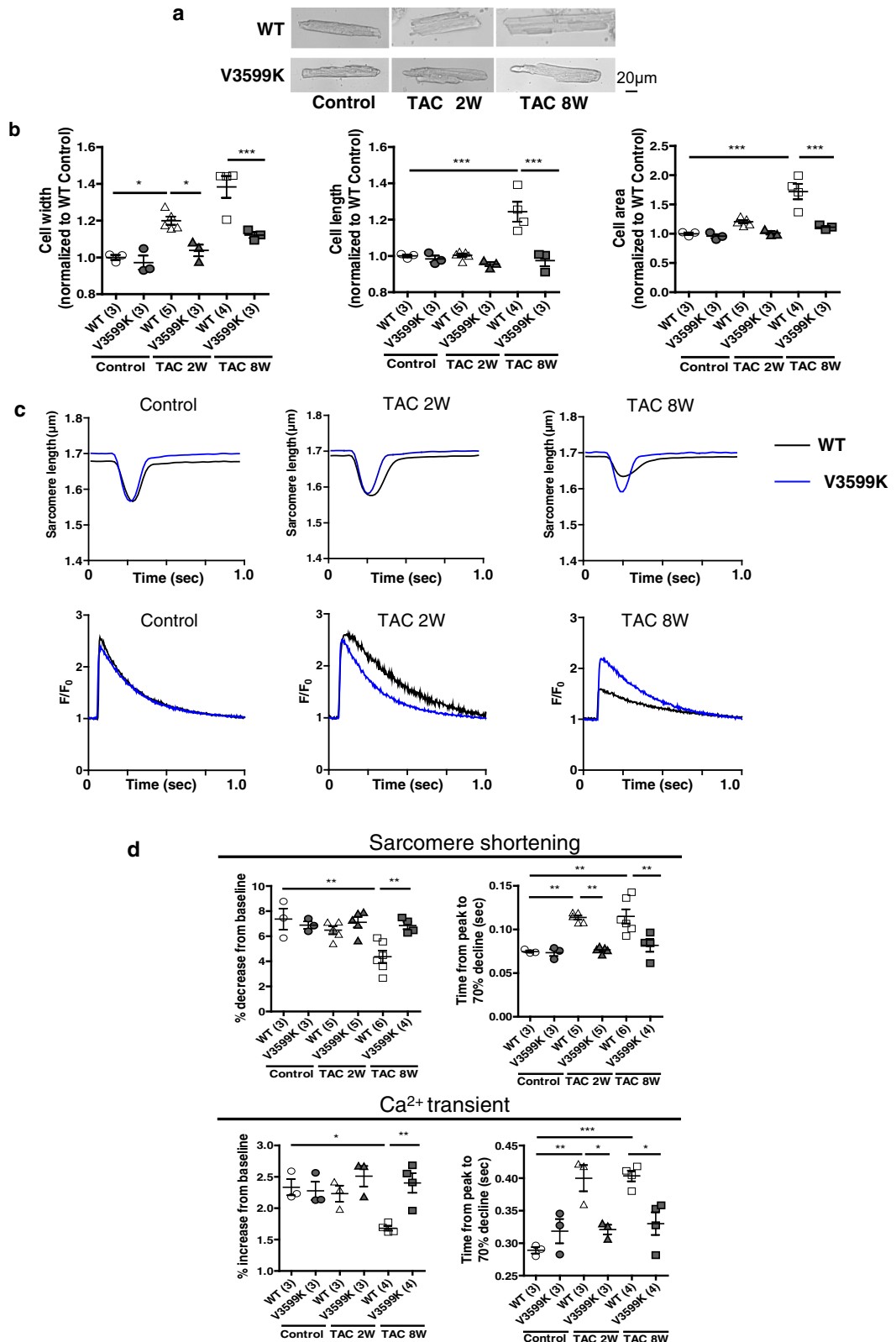

**Fig. 3 Morphology and Ca$^{2+}$ transients in intact cardiomyocytes. a** Representative images of cardiomyocytes. **b** Summarized data of cell width, cell length, and cell area in isolated cardiomyocytes. $N = 250$–400 cells from 3–5 hearts. **c** Representative recordings of sarcomere shortening and fluo-4AM fluorescence signal, at a pacing rate of 1 Hz. **d** Summarized data of sarcomere shortening, peak Ca$^{2+}$ transient, time from peak to 70% decline of Ca$^{2+}$ transient, and sarcomere shortening. $N = 22$–31 cells from 3 to 6 hearts. Values for individual mice are plotted together with mean ± SEM. Parentheses indicate the number of mice. *$P < 0.05$, **$P < 0.01$, ***$P < 0.001$ (one-way ANOVA with post-hoc Tukey's multiple comparison test).

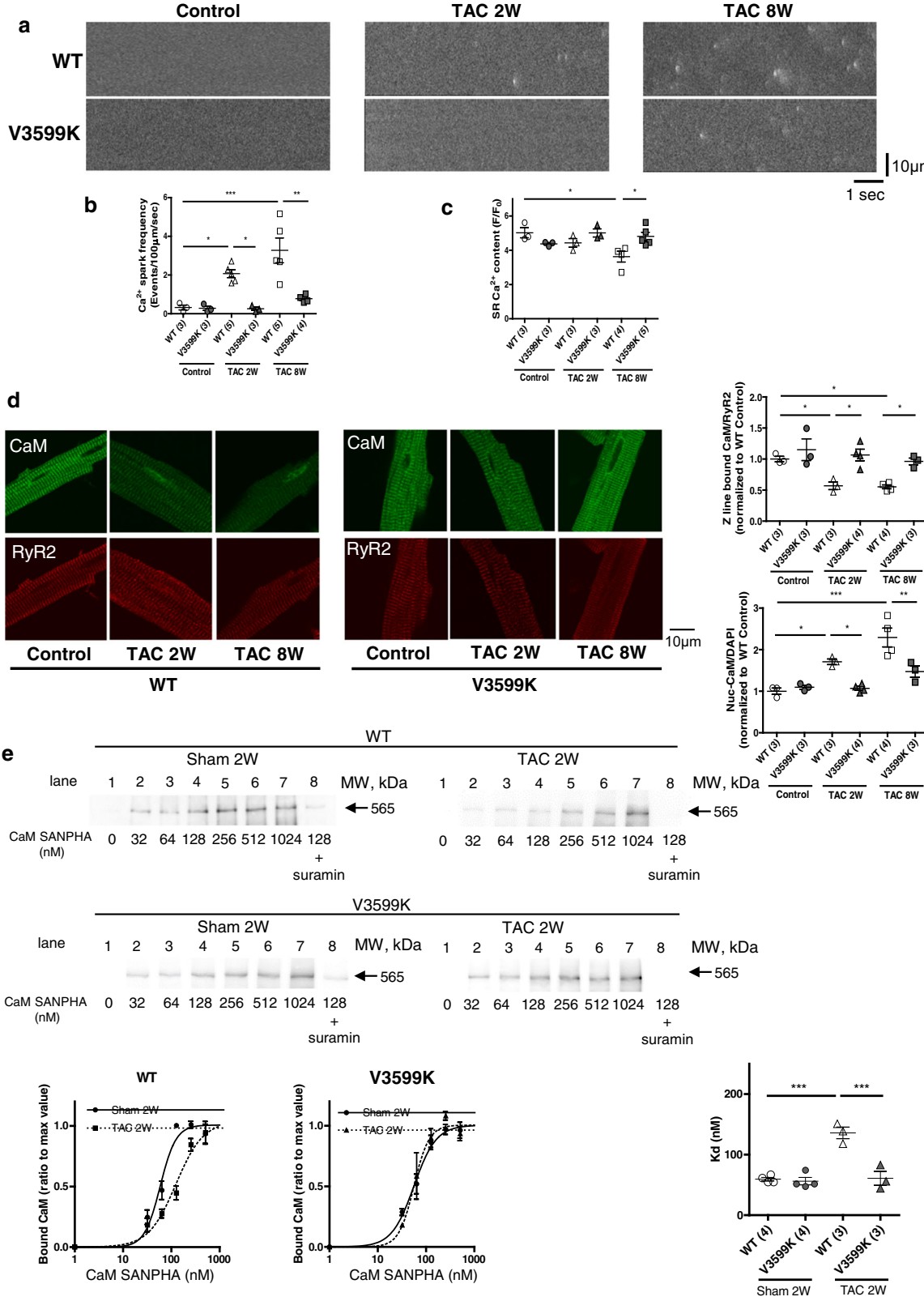

(+TAC) hearts (Fig. 5b). Actin Alpha 1 (*Acta1*), Myosin Heavy Chain 7 (*Myh7*), Natriuretic Peptide A (ANP, *Nppa*), and Natriuretic Peptide B (*Nppb*), which are known hypertrophic markers, significantly increased in WT (+TAC) hearts, compared with WT (−TAC) hearts, whereas there were no increase in V3599K (+TAC) hearts (Fig. 5c). These data were consistent with the hypertrophic morphological changes observed by

hematoxylin and eosin staining. To examine the signaling pathway that led to the hypertrophic changes in the hearts, Ingenuity Pathway Analysis (IPA) was performed using the upper 100 genes detected by factor loadings of PC3. Toxicity analysis indicated that cardiac hypertrophy was detected with a high score. In the network analysis, activation of cardiomyocytes contraction by actin-myosin complex was suggested as a top score (Fig. 5d), and

**Fig. 4 Ca²⁺ sparks and CaM-RyR2 interaction in cardiomyocytes. a** Representative recordings of spontaneous Ca²⁺ sparks. **b** Summarized data of spontaneous Ca²⁺ spark frequency. $N = 20$–$40$ cells from 3 to 5 hearts. **c** Summarized data of SR Ca²⁺ content measured from caffeine-induced Ca²⁺ transient. $N = 12$–$15$ cells from 3 to 5 hearts. **d** Representative images of endogenous CaM co-localized with RyR2 and summarized data of the Z-line bound CaM and nuclear CaM. The immuno-fluorescence signal of the Z-line bound CaM was divided by that of RyR2, normalized to control (baseline of WT), and expressed as a ratio. The immuno-fluorescence signal of nuclear CaM was divided by that of DAPI for nuclear staining, normalized to control (baseline of WT), and expressed as a ratio. $N = 20$–$38$ cells from 3 to 4 hearts. **e** (Top) Representative immunoblots of RyR2-bound CaM-SANPAH (a photoreactive crosslinker; N-succinimidyl-6-[4'-azido-2'-nitrophenylamino]). CaM binding to RyR2 was determined with immunoblotting with anti-CaM to detect RyR2-bound CaM. (Bottom) Right: summarized data of CaM binding to RyR2 as a function of the concentration of CaM–SANPAH. CaM binding was expressed as the ratio to the maximum binding of CaM (at 1024 nM). (Left) Dissociation constants (Kd). Values for individual mice are plotted together with mean ± SEM. Parentheses indicate the number of mice. *$P < 0.05$, **$P < 0.01$, ***$P < 0.001$ (one-way ANOVA with post-hoc Tukey's multiple comparison test).

the activation was indicated to be caused by intracellular Ca²⁺ released by RyR2 from data of KEGG and Gene ontology analyses (Supplementary Fig. 4a–c). *Nppa* and *Nppb* were detected in the center of the network analysis, and calcineurin, whose activity depends on Ca²⁺, was observed as an upstream molecule of *Nppa* and *Nppb* (Supplementary Fig. 5a). The signaling pathway of *Nfat* (nuclear factor of activated T cells, NFAT) family indicated as a third network was associated with ERK1 activation involving in hypertrophic change (Supplementary Fig. 5b). The upstream analysis using the gene group indicating the Ca²⁺-dependent hypertrophic pathways, detected a transcription factor myocyte enhancer factor 2 (MEF2), and the signaling pathway was suggested to participate in the induction of the hypertrophic genes such as *Acta1*, *Myh7*, and *Nppa* through GATA binding protein 4 (GATA4) activation (Supplementary Table 1 and Supplementary Fig. 6). These findings suggest that aberrant Ca²⁺ release as a result of TAC stress activates the hypertrophic signaling pathway.

**Acute systolic, but not diastolic, pressure-overload is critical for hypertrophic signaling**. Next, to clarify whether either dissociation of CaM from RyR2 or subsequent Ca²⁺ leakage is essential for the development of pressure-overload induced hypertrophy, we evaluated the effect of pressure-overload by air compression (Supplementary Fig. 7) on Ca²⁺ sparks, the ratio of CaM bound to RyR2, the export of HDAC from the nucleus to cytosol, and the import of NFAT from the cytosol to nucleus[20,21] in isolated cardiomyocytes. To eliminate the effect of contraction on hypertrophic signaling, all experiments were performed in the presence of 2, 3-butanedione monoxime (BDM) in cardiomyocytes. Hence, after electrical pacing, we could observe only Ca²⁺ transients without sarcomere shortening (Fig. 6a). Pressure-overload (150 mmHg) was applied to cardiomyocytes with 1 Hz electrical pacing during the (systolic) Ca²⁺ transient phase or (diastolic) static phase (Fig. 6a). Since myocardial stretch has been shown to cause Ca²⁺ sparks via reactive oxygen species (ROS) production[22], we anticipated that pressure-overload by air compression may also produce ROS, which may induce Ca²⁺ leakage[23]. Therefore, we measured ROS in cardiomyocytes after pressure-overload. Pressure-overload during the Ca²⁺ transient phase (but not the static phase) indeed produced ROS to a similar extent in WT and V3599K cardiomyocytes. N-acetylcysteine (NAC) (1 mM), which is an antioxidant, inhibited ROS production in WT cardiomyocytes (Fig. 6b). Furthermore, we also evaluated the effect of dantrolene on ROS, Ca²⁺ sparks, CaM-RyR2 interaction, and hypertrophic signaling in WT cardiomyocytes. Dantrolene inhibits Ca²⁺ leakage through RyR2 in CPVT or heart failure by allosterically increasing the binding affinity of CaM to RyR2[24,25]. Dantrolene did not inhibit ROS production in WT cardiomyocytes (Fig. 6b).

In WT cardiomyocytes, acute pressure-overload induced Ca²⁺ sparks, dissociation of CaM from RyR2 (Fig. 6c, d), export of

HDAC from the nucleus to cytosol, and import of NFAT from the cytosol to the nucleus, only when pressure-overload was applied during the (systolic) Ca²⁺ transient phase, but not during the (diastolic) static phase (Fig. 7). Both NAC and dantrolene inhibited all the aforementioned abnormalities, i.e., Ca²⁺ leakage and hypertrophic signaling owing to pressure-overload in WT cardiomyocytes (Figs. 6c, d, 7). In V3599K cardiomyocytes, pressure-overload did not cause Ca²⁺ sparks, CaM dissociation, HDAC, or NFAT nucleocytoplasmic shuttling (Figs. 6c, d, 7), although ROS was produced to a similar extent as in WT cardiomyocytes (Fig. 6b). These data clearly indicate that pressure-overload induced hypertrophic signaling initiates via aberrant Ca²⁺ release due to ROS-mediated CaM dissociation from RyR2.

**RyR2 destabilization directly activates hypertrophic signaling**. Because domain unzipping-mediated Ca²⁺ leakage can be caused by ROS[26], we anticipated that even without pressure-overload and subsequent ROS production, hypertrophic signaling would be induced by the direct domain unzipper, DPc10 [7]. DPc10 is a peptide that corresponds to RyR2 central domain residues 2460–2495 recapitulating CPVT (R2474S)-type arrhythmogenic RyR2 leakiness through a competing native domain with the same amino-acid residues, which associates with the corresponding N-terminal domain, hence, unzipping N-terminal (a.a. 1–600) and central (a.a. 2000–2500) domains[7]. Fluorescently labeled DPc10 was successfully delivered along Z-lines in WT cardiomyocytes by a protein delivery kit, Bioporter (Fig. 8a), whereas transport was inhibited in the presence of dantrolene, reflecting its zipping effect on the N-terminal and central interdomain interactions that may prevent DPc10 from binding to the N-terminal domain of RyR2[7]. The finding that fluorescently labeled DPc10 was not incorporated along the Z-line in V3599K cardiomyocytes (Fig. 8a) strongly supports the idea that interdomain interactions between the N-terminal and central domains are allosterically coupled with CaM-RyR2 interaction, as previously described[27], i.e., domain unzipping leads to displacement of CaM, whereas enhancement of CaM binding affinity leads to domain zipping. As expected, incorporation of DPc10 into WT cardiomyocytes (without pressure-overload) directly caused Ca²⁺ sparks, CaM dissociation from RyR2 (Fig. 8b, c), the export of HDAC from the nucleus, and import of NFAT to the nucleus, but not in V3599K cardiomyocytes (Fig. 8d, e). Taken together, our results strongly suggest that pressure-overload first produces ROS only during (systolic) Ca²⁺ transient phase, which causes oxidation of RyR2[26], in turn causing aberrant Ca²⁺ release via dissociation of CaM, thus, triggering hypertrophic signaling. Prevention of aberrant Ca²⁺ release and CaM dissociation by enhancing the affinity of CaM binding to RyR2 completely inhibits hypertrophic signaling even though ROS is produced.

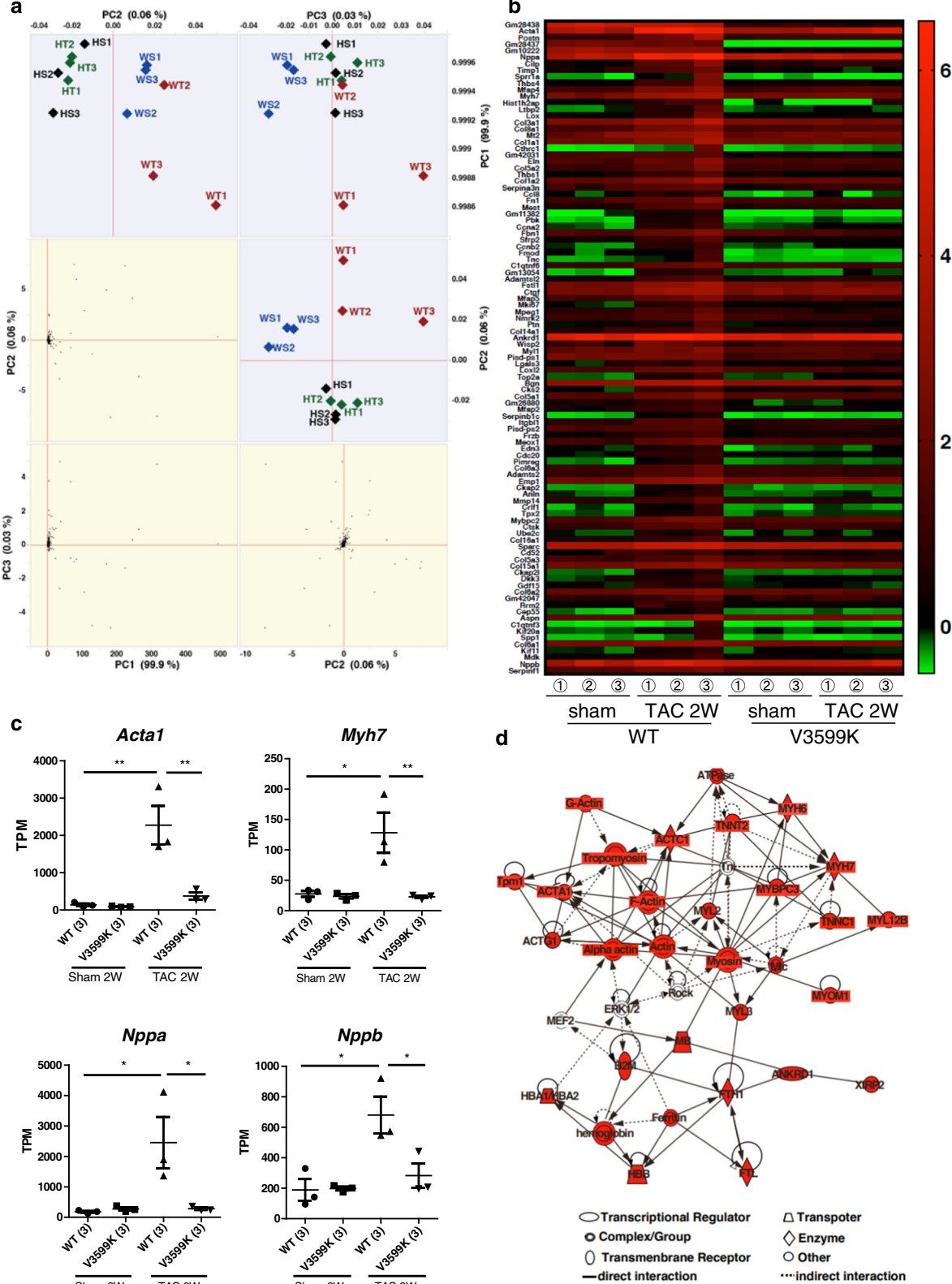

**Fig. 5 Analysis of gene expression and network pathways after chronic pressure-overload of hearts in TAC model mice. a** Principal Component Analysis (PCA) showing gene expression in WT mice {without (WS: $n = 3$) or with TAC (WT: $n = 3$) } and V3599K mice {without (HS: $n = 3$) or with TAC (HT: $n = 3$)}. **b** The heat map shows the gene expression of the upper 100 genes of scores in factor loadings of PC3. **c** The bar graphs show the gene expression of hypertrophic markers genes, *Acta1*, *Myh7*, *Nppa*, or *Nppb* in the hearts of WS, WT, HS, and HT mouse models. **d** The network shows the signaling pathway detected in upper 100 genes in factor loadings of PC3. The red symbols mean the upper 100 genes. Values for individual mice are plotted together with mean ± SEM. Parentheses indicate the number of mice. *$P < 0.05$, **$P < 0.01$ (one-way ANOVA with post-hoc Tukey's multiple comparison test).

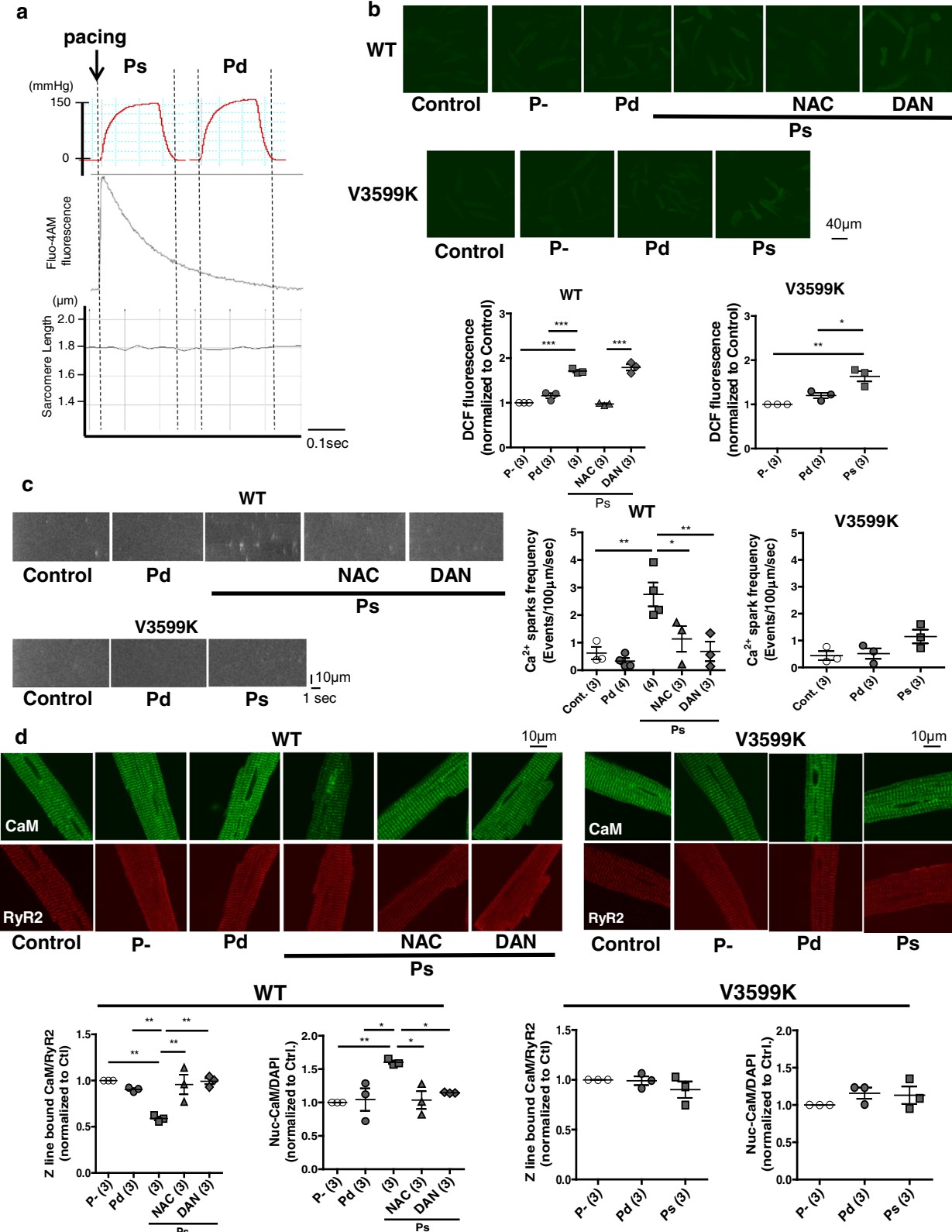

## Discussion

The major findings of this study are (1) that even in the so-called "compensatory" hypertrophic phase in which LV systolic function was intact, LV relaxation was already disturbed, whereas this was prevented by genetically increasing CaM binding affinity to RyR2; (2) that chronic inhibition of CaM dissociation prevented development of heart failure and improved prognosis; (3) that

acute pressure-overload to cardiomyocytes during the $Ca^{2+}$ transient phase, but not during the static phase, could initiate hypertrophic signaling mediated through ROS-induced CaM dissociation, even under conditions where cardiac contraction is eliminated; and (4) that enhanced binding of CaM to RyR2, followed by inhibition of $Ca^{2+}$ leakage, prevented initiation of hypertrophic signaling. To our knowledge, this is the first study to

**Fig. 6 Effect of acute pressure-overload on ROS, Ca²⁺ sparks, CaM-RyR2 interaction, and hypertrophic signaling in WT and V3599K cardiomyocytes.**
**a** Timing of acute pressure overload by air compression (+150 mmHg) to cardiomyocytes. Top: Ca²⁺ transient; bottom: sarcomere shortening. **b** Representative images of DCFHDA fluorescence and the summarized data. $N = 30–51$ cells from 3 to 4 hearts. **c** Representative images of Ca²⁺ sparks and the summarized data. $N = 19–28$ cells from 3 to 4 hearts. **d** Representative images of endogenous CaM, co-localized with RyR2, and the summarized data of the Z-line bound CaM and nuclear CaM. The immuno-fluorescence signal of the Z-line bound CaM was divided by that of RyR2, normalized to control (baseline of WT), and expressed as a ratio. The immuno-fluorescence signal of the nuclear CaM was divided by that of DAPI for nuclear staining, normalized to control (baseline of WT), and expressed as a ratio. $N = 23–39$ cells from 3 hearts. Values for individual mice are plotted together with mean ± SEM. Parentheses indicate the number of mice. *$P < 0.05$, **$P < 0.01$, ***$P < 0.001$ (one-way ANOVA with post-hoc Tukey's multiple comparison test).

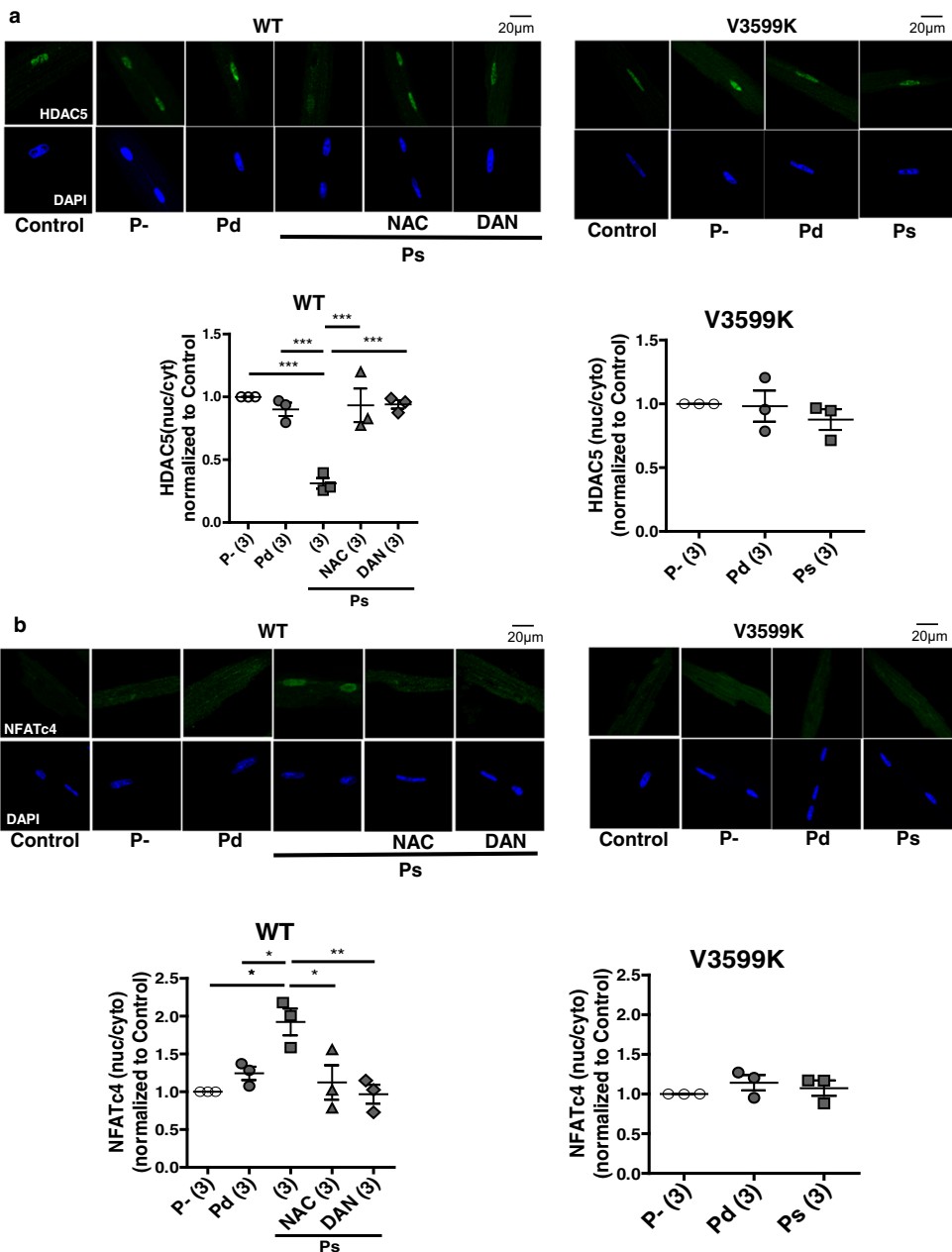

**Fig. 7 Effect of acute pressure-overload by air compression (+150 mmHg) to cardiomyocytes on hypertrophic signaling. a** Translocations of HDAC. $N = 42–60$ cells from 3 hearts. **b** Translocations of NFAT. $N = 47–58$ cells from 3 hearts Values for individual mice are plotted together with mean ± SEM. Parentheses indicate the number of mice. *$P < 0.05$, **$P < 0.01$ (one-way ANOVA with post-hoc Tukey's multiple comparison test).

demonstrate that defective interaction between CaM and RyR2 is essential to activate hypertrophic signaling owing to pressure-overload and that genetic enhancement of CaM binding to RyR2 protects against cardiac hypertrophy.

Cardiac hypertrophy is a well-known major risk factor for the development of cardiovascular diseases. Recently, accumulated evidence suggests that chronic pressure-overload induces various pathological signals, including mitochondrial dysfunction,

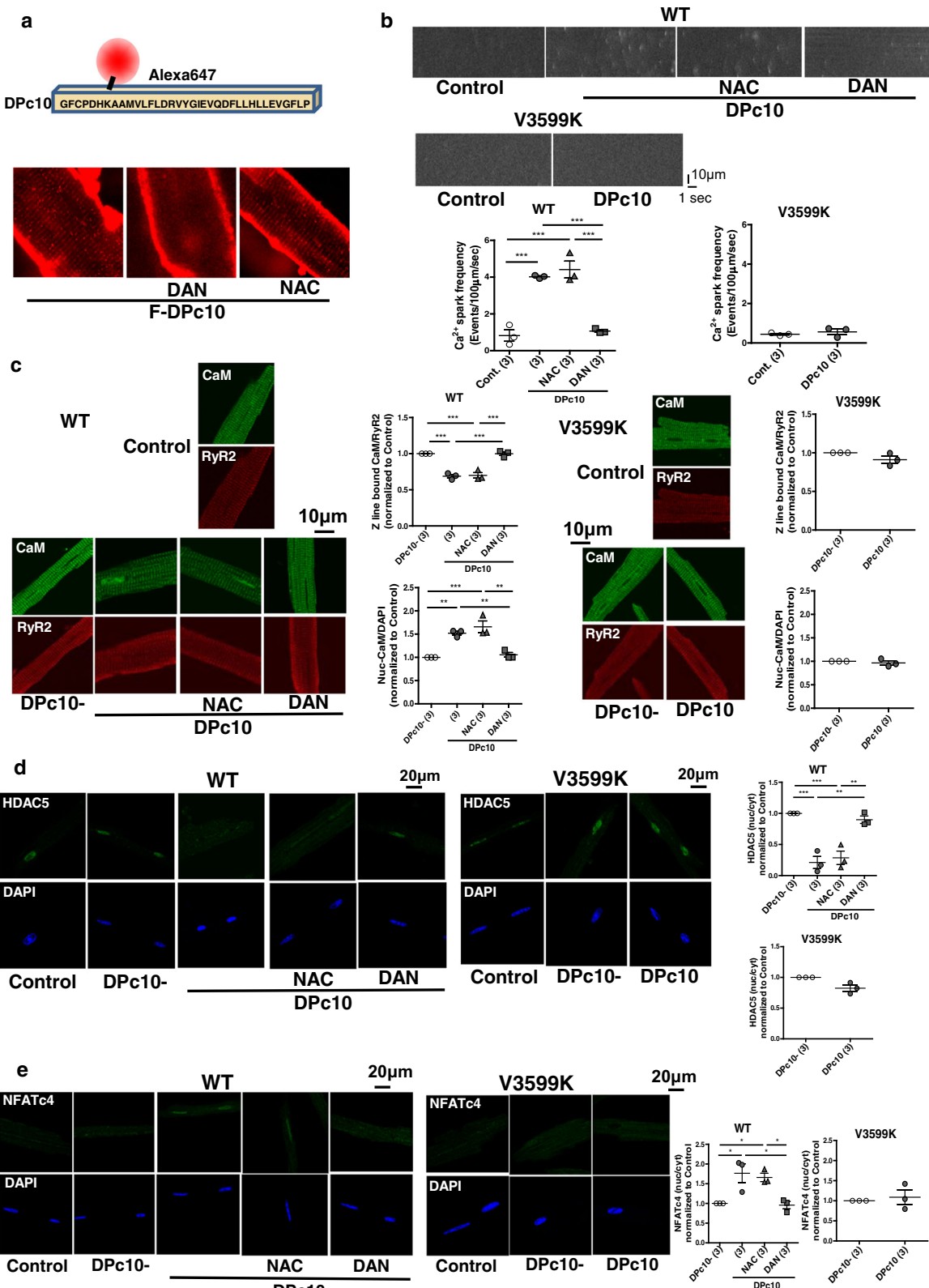

**Fig. 8 Effect of DPc10 on Ca$^{2+}$ sparks, CaM-RyR2 interaction, and hypertrophic signaling. a** Incorporation of fluorescently labeled DPc10 along the Z line in cardiomyocytes. **b** Ca$^{2+}$ sparks. $N = 18$–$31$ cells from 3 hearts. **c** Fluorescence signal of RyR2-bound endogenous CaM. $N = 19$–$25$ cells from 3 hearts. **d** Translocation of HDAC. $N = 38$–$50$ cells from 3 hearts. **e** Translocation of NFAT. $N = 36$–$53$ cells from 3 hearts. Values for individual mice are plotted together with mean ± SEM. Parentheses indicate the number of mice. *$P < 0.05$, ***$P < 0.001$ (one-way ANOVA with post-hoc Tukey's multiple comparison test).

metabolic reprogramming, impaired $Ca^{2+}$ handling, altered sarcomere structure, fibrosis, and that the adaptive response under pressure-overload has very different genetic characteristics compared to exercise and other forms of physiological hypertrophy[28]. However, it is difficult to determine whether changes in these pathological signals are the primary cause or secondary effect in many forms of hypertrophy. This study clearly demonstrated that aberrant $Ca^{2+}$ release due to CaM displacement from RyR2 is a primary cause of pressure-overload induced hypertrophy, and that prevention of $Ca^{2+}$ leak by enhancing CaM binding to RyR2 completely inhibits hypertrophy. Moreover, based on the findings that V3599K mice showed much better cardiac function and prognosis after TAC than WT mice, it is strongly suggested that cardiac hypertrophy is indeed a maladaptive response from the beginning and should be corrected as early as possible after pressure-overload is applied to cardiac muscle.

The mechanism by which V3599K RyR2 mutated hearts adapt to chronic pressure-overload without hypertrophy needs to be explained. The Ees of LV pressure-volume relationship was steeper after TAC in V3599K mice than in Sham (V3599K) mice, indicating increased contractility (Fig. 2). This increased contractility may explain why the V3599K heart can adapt to chronic pressure-overload without stretching. However, it is unclear why LV contractility increases in the V3599K hearts. The signaling pathways antagonizing pathological hypertrophic responses, generally observed in "physiological hypertrophy", might be involved in the increase in LV contractility; e.g., cell survival signaling, increased energy production and efficiency, antioxidant systems, mitochondrial quality control, and cardiomyocyte proliferation and regeneration[28]. In particular, given that there was no cellular hypertrophy, and there was no difference in sarcomeric shortening before and after TAC in V3599K cardiomyocytes, cell proliferation processes might be involved in the increase in LV contractility of V3599K hearts.

To elucidate the precise mechanism of hypertrophic signaling after pressure-overload, we have developed a chamber system in which air compressive pressure-overload can be imposed on cardiomyocytes during an arbitrary period after electrical pacing, and we have obtained several novel findings. First, pressure-overload imposed only during the $Ca^{2+}$ transient phase, but not during the static phase, activated hypertrophic signaling, suggesting that cytosolic increase in $Ca^{2+}$ is a prerequisite for pressure-overload induced hypertrophy. Second, pressure-overload to cardiomyocytes first produced ROS, subsequently leading to CaM dissociation from RyR2 presumably owing to the oxidation of RyR2[26], causing $Ca^{2+}$ leakage and activation of hypertrophic signaling. Third, DPc10, an unzipper of RyR2, caused CaM dissociation from RyR2, which in turn induced $Ca^{2+}$ sparks without increasing ROS, thereby activating hypertrophic signaling. Fourth, genetic enhancement of CaM binding to RyR2 inhibited hypertrophic signaling induced by either pressure-overload or DPc10. Taken together, aberrant interactions between calmodulin and RyR2 consist a critical initial trigger for the development of pressure-overload hypertrophy. Consistent with this hypothesis, our whole transcriptome analysis demonstrated that hypertrophy-related genes including *Acta1*, *Myh7*, *Nppa*, and *Nppb* did not increase after TAC when dissociation of CaM was genetically prevented.

Inter-domain cross-talk between the zipping interface and CaM binding site was based on the reciprocal nature of CaM binding and DPc10 binding to RyR2. Namely, exposing RyR2 to DPc10 inhibits CaM binding to RyR2 and CaM binding to RyR2 reduces DPc10 accessibility to its site[27]. Gong et al. more recently reported that Apo-CaM and $Ca^{2+}$-CaM bind to distinct but overlapping sites in an elongated cleft formed by the helical, handle, and central domains in RyR2 (PDB ID: 6JV2 also see

ref. [15]). They found that the N- and C-lobes of $Ca^{2+}$-CaM interact with the C- (3596–3605) and N termini of helix $\alpha-1$ (3587–3592), respectively, and N-lobe of $Ca^{2+}$-CaM also interacts with Arg2209 in helix 2b of handle domain and Met3820 in helix $\alpha-9$. According to their cryo-EM structures of RyR2, the "zipping interface" between N-terminal and central domains appears to be smaller than we have expected, and also it is located at the inter-subunit interface of RyR2, which is close to CaM binding region (Supplementary Fig. 8). Therefore, the "core domain unzipping" between N-terminal (1–220) and central (2250–2500) domains may cause defective inter-subunit interaction, allosterically induce the conformational change at CaM binding region, thereby displacing CaM in failing hearts. Very interestingly, the binding site of dantrolene (601–620)[25] is just adjacent to that of another RyR2 stabilizer, K201 (JTV519), which we first discovered a stabilizing effect on the RyR2 channel[29] and identified the binding site (2114–2149)[9], suggesting that both drugs commonly correct defective CaM-RyR2 interactions in diseased hearts (Supplementary Fig. 8).

The mechanism by which the V3599K mutation prevents the reduced affinity of CaM for binding to RyR2 in pressure-overloaded hearts still remains to be elucidated. Given that the affinity of CaM binding to the V3599K-mutant RyR2 was similar to WT RyR2 unless pressure-overload was imposed (Fig. 4e), CaM accessibility to the V3599K-mutated CaM binding site may be allosterically enhanced, only when the defective inter-subunit interaction of RyR2 is produced. Since CaM interacts densely with three domains (helix $\alpha-1$, helix 2b, helix $\alpha-9$) in RyR2[15], in situ CaM accessibility to its binding site may not be easily enhanced by the V3599K mutation, though the V3599K mutation in the CaM binding domain peptide (3583–3603) markedly enhanced the binding affinity to CaM[17]. In contrast, the defective inter-subunit interaction due to pressure-overload may allosterically change the conformational state of the CaM binding region, facilitating CaM binding to the V3599K mutant (not WT) domain in helix $\alpha-1$. The fact that the CaM binding region is three-dimensionally close to the "zipping interface" supports this idea (Supplementary Fig. 8).

There are two major $Ca^{2+}$-activated cardiac hypertrophy signaling pathways, the $Ca^{2+}$/calmodulin-dependent protein kinase II (CaMKII)- HDAC pathway and the CnA- NFAT pathway[20,21]. CaMKII-mediated phosphorylation of HDAC results to its export from the nucleus, leading to relief of HDAC-mediated transcriptional repression of MEF2 and MEF2-dependent transcription of hypertrophic genes[20]. In addition, CnA dephosphorylates NFAT, which leads to nuclear translocation of NFAT and upregulation of NFAT-dependent transcription of hypertrophic genes such as ANP and GATA4[21]. Consistent with the previous reports, MEF2C was detected as an upstream factor in the pathway analysis of genes activated in TAC model, and the activation was indicated to be mediated by GATA4 activation and HDAC inhibition. Of particular interest, here we showed that pressure-overload mediated de-stabilization of RyR2 provides two essential factors required for these two major hypertrophic signaling pathways, namely, CaM and diastolic increase in $Ca^{2+}$, at the same time. In support of this notion, we demonstrated that acute pressure-overload imposed on cardiomyocytes induces export of HDAC from the nucleus and import of NFAT to the nucleus, via dissociation of CaM from RyR2, and that either pharmacological or genetic rescue of reduced CaM binding to RyR2 prevented translocation of HDAC and NFAT (Fig. 9).

Heart failure with preserved ejection fraction (HFPEF) is the most common form of heart failure in the elderly population, characterized by exercise intolerance, poor quality of life, frequent hospitalizations, and reduced survival[30]. Although diastolic dysfunction with cardiac hypertrophy is an important determinant

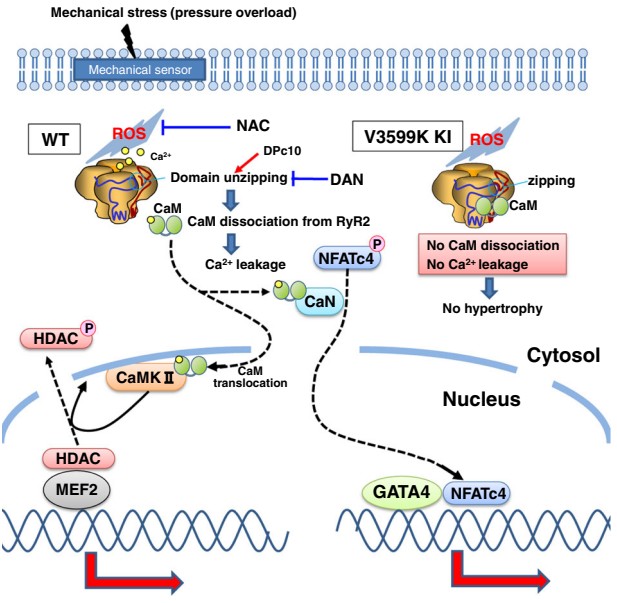

**Fig. 9 Molecular mechanism of pressure-overload induced cardiac hypertrophy.** Pressure-overload first induces ROS production, which in turn induces CaM dissociation and Ca2+ leakage, thereby activating hypertrophic signaling mediated through the CaM-CaMKII and Ca$^{2+}$-calcineurin pathway. V3599K mutation within CaM binding domain of RyR2 enhances CaM binding to RyR2, thereby inhibiting CaM dissociation and Ca$^{2+}$ leakage despite ROS production due to pressure-overload, and thus hypertrophic signaling is not activated.

for the pathogenesis of HFPEF, all possible therapeutic candidates to inhibit cardiac hypertrophy, such as inhibitors of the renin-angiotensin-aldosterone system, i.e., ACE inhibitor, AngII antagonists, and aldosterone inhibitors, failed to improve prognosis of patients with HFPEF[30]. Therefore, the optimal therapy still remains uncertain and an effective therapeutic option for HFPEF is urgently required. The inhibitory action of dantrolene against hypertrophy is mediated through direct stabilization of RyR2, and hence, it may be effective to any type of hypertrophy triggered by a variety of factors, as long as aberrant Ca$^{2+}$ release is involved. Therefore, the inhibitory action of dantrolene against hypertrophy may be a new candidate for HFPEF treatment.

In conclusion, cardiac hypertrophy/heart failure induced by pressure overload is caused by ROS-mediated destabilization of RyR2 (defective inter-domain interaction→CaM dissociation→ Ca$^{2+}$ leakage), and by increasing the binding affinity of CaM to RyR2 pharmacologically or genetically, the abnormal pathways leading to hypertrophy are inhibited, thereby suppressing the progression to heart failure.

## Methods

**Mice.** Ten- to 12-week-old C57BL/6 mice (male or female) were used in this study. WT C57BL/6 mice were obtained from Japan SLC, Inc. (Hamamatsu, Japan). V3599K mice were made by UNITECH Co., Ltd (Chiba, Japan). All in vivo experiments were performed in contemporary random order. Allocation concealment was not applicable because the genetic background of each experimental group was different, and this study did not involve any testing of therapeutic agents. Measurement and analysis of the acquired data were performed without the investigators knowing any genetic information of the mice. This study conformed to the Guide for the Care and Use of Laboratory Animals published by the US National Institutes of Health (NIH Publication No. 85–23, revised 1996). All experimental protocols were approved by the Animal Ethics Committee of Yamaguchi University School of Medicine. The care of the animals and the protocols used were in accordance with the guidelines from the Animal Ethics Committee of Yamaguchi University School of Medicine.

**Transthoracic Echocardiography.** Echocardiograms were obtained, as described previously[10]. Cardiac function was analyzed by an F37 ultrasound machine (Hitachi medical) equipped with a 7.5-MHz probe (Hitachi, UST-5413). Mice were initially anesthetized with 4 to 5% isoflurane (mixed with oxygen) and maintained with 1 to 2% isoflurane during echocardiography. Cardiac ventricular dimensions were measured on M-mode echocardiograms.

**Hemodynamic analysis.** Mice were anesthetized with 1.5% isoflurane and placed on a heating pad (37 °C). A 1.4-F high-fidelity micromanometer catheter (Millar Instruments, Houston, TX) was introduced into the right carotid artery and advanced across the aortic valve into the left ventricular cavity. To assess a LV pressure-volume relationship over a wide range of pressure, occlusion of inferior vena cava (IVC) was performed. LV volume was calculated by Teichholz's formula: LV Volume = [7/(2.4 + LVID)] * LVID[3]; ID = internal diameter. Ees was calculated as the slope of end-systolic pressure-volume relationship during IVC occlusion.

**Isolation of cardiomyocytes.** Cardiomyocytes were isolated from mouse hearts as described previously[7]. Briefly, mice were anesthetized with pentobarbital sodium (70 mg/kg of body weight, i.p.), intubated, and ventilated with ambient air. An incision was made in the chest, and the heart was quickly removed and retrogradely perfused with collagenase-free buffer via the aorta under constant flow. The left ventricular myocardium was minced with scissors in the fresh collagenase-containing buffer and rod-shaped adult mouse cardiomyocytes were prepared with retrograde perfusion of the hearts with 95% O$_2$ and 5% CO$_2$-bubbled minimal essential medium (Sigma, St Louis, MO, USA) supplemented with 50 µmol/L [Ca$^{2+}$], 0.5 mg/mL collagenase B, 0.5 mg/mL collagenase D, and 0.02 mg/mL protease type XIV. Ca$^{2+}$ concentration was then gradually increased to a final concentration of 1 mmol/L by changing the incubation medium (50, 100, 300, 600, and then 1 mmol/L). Isolated mouse cardiomyocytes were transferred to laminin-coated glass culture dishes and incubated for several hours at 37 °C in a 5% CO$_2$ and 95% O$_2$ atmosphere.

**Monitoring of Ca$^{2+}$ transients of cardiomyocytes.** Isolated ventricular myocytes were incubated with 20 µmol/L fluo-4 AM for 30 min at 24 °C and washed twice with Tyrode's solution, as described previously[7]. All experiments were conducted at 35 °C. Intracellular Ca$^{2+}$ measurements of cells stimulated by a field electric stimulator (Ion Optix, MA, USA) were acquired using a fluorescent digital microscope (BZ9000, Keyence, Japan).

**Analysis of Ca$^{2+}$ sparks and SR Ca$^{2+}$ content.** Ca$^{2+}$ sparks were measured as previously described[7] using a laser scanning confocal microscope (LSM-510, Carl Zeiss) equipped with an argon-ion laser coupled to an inverted microscope (Axiovert 100, Carl Zeiss) with a Zeiss 40X oil-immersion Plan-Neofluor objective (numerical aperture, 1.3; excitation at 488 nm; emission at > 505 nm). Briefly, intact cardiomyocytes were loaded with fluo-4 AM (20 µmol/L; Molecular Probes) for 30 min at 24 °C. Line-scan mode was used, where single cardiomyocytes were scanned repeatedly along a line parallel to the longitudinal axis, avoiding the nuclei. To monitor Ca$^{2+}$ sparks, cardiomyocytes were stimulated until the Ca$^{2+}$ transient reached a steady state. Stimulation was then stopped, and Ca$^{2+}$ sparks were recorded during the subsequent ~10 s rest. Data were analyzed with SparkMaster, an automated analysis program that allows for rapid and reliable spark analysis[31]. The variables analyzed included general image parameters (like a number of detected sparks and spark frequency) as well as individual spark parameters (amplitude, full width at half maximum [FWHM], and full duration at half maximum [FDHM]). To assess SR Ca$^{2+}$ content, caffeine (10 mmol/L) was rapidly perfused to discharge SR-loaded Ca$^{2+}$.

**Immunocytochemistry analysis.** Antibodies against the following proteins were used for immunohistochemistry: RyR2 (C3–33; Sigma-Aldrich, St. Louis, MO, USA), CaM (EP799Y, Abcam, Cambridge, UK), HDAC5 ((NBP2-22152, Novus Biologicals), NFATc4 (ab3447, Abcam, Cambridge, UK).

**Immunocytochemistry analysis of endogenous RyR2-bound CaM.** Isolated cardiomyocytes were fixed with 2% paraformaldehyde for 2 min and permeabilized with methanol at −20 °C, as described previously[11]. Then, cardiomyocytes were incubated overnight at 4 °C with anti-CaM and anti-RyR in 1% bovine serum albumin and 0.5% Triton X-100, followed by labeling with an Alexa488-conjugated goat anti-rabbit and an Alexa633-conjugated goat anti-mouse secondary antibody. The sarcomere-related periodical increase in Alexa633 and Alexa488 fluorescence intensity from baseline was integrated with respect to the longitudinally selected distance (~25 µm), and then the value was divided by it. The mean value of one sarcomere-related increased fluorescence intensity was then calculated as the arbitrary amount of RyR and RyR-bound CaM.

**Analysis of association of CaM to RyR2 by the CaM-SANPAH crosslinking method.** The binding of CaM to RyR2 was evaluated using the photoreactive cross-linker, sulfosuccinimidyl-6-[4′-azido-2′-nitrophenylamino]hexanoate

(Sulfo-SANPAH, Thermo Fisher Scientific, Waltham, MA), as described previously[11]. First, a CaM-SANPAH conjugate was prepared by incubating 50 μmol/L recombinant CaM in conjugation buffer (150 mmol/L KCl and 20 mmol/L MOPS at pH 7.2) and 100 μmol/L sulfo-SANPAH in the dark for 30 min. Conjugation was quenched by adding an excess amount of lysine. CaM-SANPAH was purified using an Amicon Ultra filter (MWCO 10k). Mouse cardiac homogenates were diluted with binding buffer (150 mmol/L KCl, 10 μmol/L CaCl$_2$, and 20 mmol/L MES at pH 6.8) to 1 mg/mL and mixed with 100 nmol/L CaM-SANPAH conjugate in the dark in a glass tube. After a 10 min binding time, UV-crosslinking was applied and the sample buffer was added to crosslinked SR membranes followed by western blotting using anti-CaM (Merck, Millipore, Darmstadt, Germany). CaM-SANPAH crosslinked to RyR2 was detected as a 550 kDa band.

**Acute pressure-overload to isolated cardiomyocytes.** Supplementary Fig. 7 shows the scheme and detailed description of the acute pressure overload system. Pressure-overload (150 mmHg, 250 ms) was applied to cardiomyocytes with 1 Hz electrical pacing during the (systolic) Ca$^{2+}$ transient phase or (diastolic) static phase. Ca$^{2+}$ sparks were measured after 10 min of pressure overload. Endogenous CaM-RyR2 binding experiments and hypertrophic signaling assay (HDAC, NFAT) were performed after 30 min of pressure overload. Data were recorded at each assay using 10 to 20 cardiomyocytes per dish.

**Histology.** Hearts from 12–20 weeks old WT and V3599K mice were fixed using 10% formalin. A complete, full-circumferential section, at the level of the two left ventricular papillary muscles, was selected for morphometric analysis. H&E and Masson's trichrome staining were performed for each section of the ventricle.

**Whole transcriptome analysis with RNA-seq.** Total RNA was extracted from the hearts of mice using the RNeasy Mini Kit (Qiagen), and mRNA was purified with oligo dT beads (NEBNext Poly (A) mRNA magnet Isolation Module, New England Biolabs, NEB). Complementary DNA (cDNA) libraries for Illumina sequencing were generated with NEBNext Ultra II RNA library Prep kit (NEB) and NEB-Nextplex Oligos for Illumina. Briefly, fragmentation was performed with NEBNext Random Primers in NEBNext First Strand Synthesis Reaction Buffer at 94 °C for 15 min, and reverse transcription was performed with NEBNext Strand Synthesis Enzyme Mix. After ligation, library fragments were enriched, and index sequences were inserted with PCR amplification. Following purification with AMPure XP beads (Beckman Coulter), the quality and the concentration of the libraries was evaluated using an Agilent 2200 TapeStation (D1000, Agilent Thermo Fisher). The confirmed libraries were mixed to equal molecular amounts against the clustering and sequencing on an Illumina Next-seq DNA sequencer with 75 bp pair-end cycle sequencing kit (Illumina). To produce the raq bcl, or base call files, quality assessment and image analyses were performed using Next-seq packaging software Real Time Analysis, and bcl2fastaq Conversion Software v2.19 was used for de-multiplexing of the samples. Reads with more than two ambiguous nucleotides and reads with quality scores less than 20 as calculated by the Phred program were removed using CLC Genomics Workbench software (ver.8.01, Qiagen). Long reads with more than 1000 nucleotides and short reads with fewer than 20 nucleotides were also discarded. Trimmed reads were mapped to the mouse reference genome GRCm38 release-92 in default settings. Briefly, reads were aligned to reference sequences using the setting conditions with mismatch cost of 2, insertion cost of 3, and deletion cost of 3. In addition, reads were mapped when at least half of the alignment matched the reference sequence (length fraction of 0.8), and the matched alignment was at least 80% identical to the reference sequences (similarity fraction of 0.8), and non-specifically matched reads within the 10 regions were mapped randomly. For pathway analysis, factor loadings were calculated with Principal Component Analysis of JMP Pro ver.14.0 software, the upper 100 genes were selected, and the pathway analysis for the detected genes was examined using Ingenuity Pathway Analysis (Qiagen), KEGG pathway analysis (http://www.kegg.jp/ or http://www.genome.jp/kegg/), and GO analysis (http://www.geneontology.org).

**Statistics and Reproducibility.** Unpaired t-tests were used for statistical comparisons of data obtained during two different conditions, whereas ANOVA with a post hoc Tukey's test was used for statistical comparison of more than two groups. All data are expressed as the means ± SE. A probability value of less than 0.05 was considered statistically significant. Each experiment was repeated at least three times, and sample sizes and numbers are indicated in detail in each figure legend.

**Reporting Summary.** Further information on research design is available in the Nature Research Reporting Summary linked to this article.

## Data availability
The authors declare that all data supporting the findings of this study are available within the paper and its Supplementary Information files. The source data underlying all figures (Figs. 1–9, Supplementary Fig. 1–8, Supplementary Table 1) are provided as Supplementary Data 1 and Supplementary Data 2. In addition, RNA-seq data are deposited in Gene expression omnibus (GEO) (GEO accession no.: GSE158536).

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

## Acknowledgements
This work was supported by grants-in-aid for scientific research from The Ministry of Education in Japan (Grant Nos. 26293189, 20H03677 to M.Y., 15K09085 to T.Y., 15K09142 to S.K.). We wish to acknowledge Yoko Okamoto, Satomi Tateda for their technical assistance.

## Author contributions
M.K., S.K., and T.Y. designed the research, performed experiments, analyzed data, and wrote the manuscript. R.Y., T.K., S.F., Y.N., T.K., H.U., T.O., and S.O. assisted the measurement of $Ca^{2+}$ transients, cell shortening, and $Ca^{2+}$ sparks and data analysis thereof. K.W. and Y.M. performed whole transcriptome analysis using RNA-seq analysis. M.Y. designed and supervised the research and wrote the manuscript.

## Competing interests
The authors declare no competing interests.
