## [Peer Review File · Communications Biology]

Reviewers' Comments:

Reviewer #1:

Remarks to the Author:

The present study by Yano and colleagues aimed to address an interesting question about calcium leak from RyR2 during pressure overload-induced hypertrophic growth and heart failure. The authors convincingly showed that the unzipping of RyR2 even in the early phase of hypertrophic growth may lead to decompensation and precipitation of cardiomyopathy. Importantly, using a genetic mouse model, the authors could enhance RyR2-CaM interaction and successfully prevent heart failure under pressure overload. The authors went further to show that the underlying mechanisms include ROS (in systolic phase) and modification of RyR2. Overall, this is a well-designed study and the conclusions are strongly supported by the data. A few concerns are listed as the following.

1. If the unzipping of RyR2 and calcium leaking is a critical mechanism of pressure overload-induced heart failure, enhancing the interaction between RyR2 and CaM would be cardioprotective. Have the authors tried to use Dantrolene as a therapeutic agent either before pressure overload (prevention) or after the establishment of heart failure (rescue)? To make it more convincing, Dantrolene treatment in RyR2 V3599 mutant mice would not provide additional improvements.
2. GO or KEGG analysis in Figure 4 would be helpful to understand molecular changes in the transgenic heart after TAC.
3. I recommend moving the schematic graph in Supp. Figure 4 to main figures.

Reviewer #2:

Remarks to the Author:

This paper by Kohno et al., is a comprehensive and detailed biochemical-based approach to investigate the in vivo and cellular phenotype of the homozygous RyR2 V3599K mutant mouse strain they generated.

There is a considerable depth of experiments, with accompanying quality of work performed. Overall, this is a very nice manuscript that is a natural follow-up from other published work from this research group investigating the role of RyR/CaM interactions in the heart. However, there are a few issues in the paper that could be improved and clarified.

1. The 2 wk TAC (?hypertrophy) data in WT animals is very mild, with only a modest HW/BW change. Clarifying this mild response might be appropriate. The 8wk data is quite clear and is not a concern.
2. Fig 2C F/Fo for baseline and 2wk TAC.. Is there shift in time to activation? or is this simply an alignment issue. The upstroke of the transient is offset in the mutant. Clarification of this difference would improve the interpretation of this mutant.
3. For all the imaging-based experiments, I was surprised that there were no negative controls or images for any of the studies. Given that the data are presented as colocalization between molecules (Cam/RyR; or cyto/nuc) negative controls with their data are needed. This would be for fig3d, 5b, 5d, 6 and 7.
4. the RNAseq data brings a novel approach to all of the biochemistry and imaging studies. however as presented it is not very well described or insightful. For instance, why not provide targeted transcriptomic analysis for all myosins, gata's, NFAT, nkx members, RyR1-3, CAM, CAMKII, etc. As presented a network confirming ANF is up, is not worth an entire figure panel.

Minor issues:

1. in the title of the first two paragraphs (line 102 and line 118), there is no actual data showing 'Enhanced caM binding to RyR2...'. I believe the authors are inferring from their previous paper,

and these titles could be edited to reflect what is actually in figures 1 and 2.

2. There are only a few grammatical/stylistic errors in the paper, which would likely be picked up by a typesetting team. For instance, line 320, what does 'it' refer to; the inhibitory action of dantrolene?

3. Add molecular weights for bands in Fig 3e.

4. The last paragraph, for this reviewer, was 'wordy' and did not express the manuscript clearly.

Reviewer #3:

Remarks to the Author:

In this manuscript, Kohno et. al. study a recently-characterized mouse model featuring enhanced RyR2 affinity for apoCaM (V3599K) in the setting of thoracic aortic constriction-induced pathological hypertrophy and failure. They find that the V3599K mouse has a blunted pathological response, both in cardiac morphometry as well as in downstream calcium signaling, to hypertrophic stimuli and that this leads to enhanced survival of the animals. The paper is fairly strong in the data presented. There are some methodological concerns, and the manuscript could be updated based on literature more current than that cited by the authors, but overall this was an insightful submission.

Given the coronavirus pandemic, I hope the authors are safe and the comments below helpful without being overly cumbersome experimentally.

1. It is unclear if the V3599K effect is entirely due to enhanced CaM binding. In Fig. 3E, the affinity appears very similar between WT and V3599K. The evaluation of affinity is based on peptide-binding studies from the groups prior JCI Insight manuscript. I don't see any data that this holds true in vivo. The results are certainly interesting, but are there other structural changes that this mutation might engender that are responsible for these effects?

2. Along the same lines, I would suggest that the results be contextualized in the setting of the substantial structural data from Nieng Yan's group that has become available on RyR2, including the CaM-bound structures (Gong et. al. Nature 2019, 572: 347). It is surprising the authors neither discuss nor cite this recent data. A figure should be included demonstrating where the V3599K mutation is in relation to CaM binding. The zipping/unzipping concept is simple but outdated, and the manuscript should be substantially rewritten placing the results within the context of the newer structural data.

3. I find it confusing that the authors repeatedly claim that the V3599K model shows that compensatory hypertrophy is not physiological but pathological. I think this is fairly well established. The citations used by the authors are several decades old. More recent literature (e.g. reviewed in Nat Rev Cardiol. 2018 Jul;15(7):387) show that the adaptive responses under pressure overload have very different genetic signatures compared to exercise or other forms of "physiological" hypertrophy. I am not entirely sure what new point the authors are trying to make here.

4. The authors see neither physiological nor pathological hypertrophy following TAC (though if you look carefully at Fig. 1B and 2B there may be a very mild, blunted response). If there is not substantial cellular or organ level hypertrophy, how do these hearts generate the same peak pressures? Is it a change in Frank-Starling mechanism? Was TAC not as effective in the transgenic mice? There needs to be some physical mechanism to handle the increased afterload induced by TAC, but this is neither discussed nor presented in the data here. It would be beneficial to include some organ, cellular, or molecular data to suggest how the transgenic hearts adapt to the increased afterload.

5. The data in Fig. 5 on direct pressure load on cardiomyocytes is intriguing but hard to interpret. I couldn't find a detailed description, citation, or validation of the "air compression" system. I'm not sure how this is supposed to mimic pressure overload and though the results are consistent with their hypothesis, the entire experiment is confusing and I couldn't really place any confidence in it.

6. The survival data for sham V3599K in Fig. 1E is missing. Along these same lines, I wondered if there was some underlying pathology induced at baseline by the V3599K mutation? The authors might speculate why this mutation has not occurred spontaneously if its effects are largely beneficial?
7. Contrast in a lot of the images is poor (e.g. Fig 5B) and scale bars are missing in most.
8. The speculation on the usefulness of dantrolene was welcome. It certainly seems like an intriguing option for a potential subset of HF patients.

Dear Editor,

We very much appreciate valuable comments provided by you and the reviewers. In response to these comments we have made extensive revisions. Below are point-by-point descriptions of the revisions we have made, which, I hope, make this manuscript suitable for publication. For the reviewers' convenience, the revised portions are underlined in the text.

Responses to the reviewer #1

Thank you very much for valuable comments.

1. If the unzipping of RyR2 and calcium leaking is a critical mechanism of pressure overload-induced heart failure, enhancing the interaction between RyR2 and CaM would be cardioprotective. Have the authors tried to use Dantrolene as a therapeutic agent either before pressure overload (prevention) or after the establishment of heart failure (rescue)? To make it more convincing, Dantrolene treatment in RyR2 V3599 mutant mice would not provide additional improvements.

We recently reported that chronic treatment with dantrolene improved cardiac and cellular functions even after cardiac dysfunction and LV remodeling had already developed in CaMKII δ c transgenic (TG) mice that show defective CaM binding to RyR2 (Biochem Biophys Res Commun. 2020 Apr 2;524(2):431-438). Therefore, we anticipate that dantrolene would ameliorate cardiac and cellular functions even after heart failure developed in WT TAC mice as well.

We anticipate that dantrolene treatment for V3599K mutant mice would not provide any additional improvements as suggested by the reviewer, because either dantrolene or V3599K mutation commonly mediates the increase in CaM binding affinity to RyR2. Because time has been limited for revision, we could not perform *in vivo* experiments. Thank you very much for your valuable comments.

2. GO or KEGG analysis in Figure 4 would be helpful to understand molecular changes in the transgenic heart after TAC.

Following the suggestion by the reviewer#1, we carried out KEGG pathway and Gene

ontology (GO) analyses. KEGG pathway suggests that intracellular Ca^{2+} released by RyR2 activated the cardiac muscles contraction through actin-myosin complex.

We inserted supplementary figure and the related sentences. In the network analysis, activation of cardiomyocytes contraction by actin-myosin complex was suggested as a top score (Fig. 5D), and the activation was indicated to be caused by intracellular Ca^{2+} released by RyR2 from data of KEGG and Gene ontology analyses (supplementary Fig. 4A-C).

3. I recommend moving the schematic graph in Supp. Figure 4 to main figures.

We moved the schematic illustration to main Figure 9.

Responses to the reviewer #2

Thank you very much for valuable comments.

1. The 2 wk TAC (?hypertrophy) data in WT animals is very mild, with only a modest HW/BW change. Clarifying this mild response might be appropriate. The 8wk data is quite clear and is not a concern.

As pointed out by the reviewer, the extent of hypertrophy was indeed modest at either cardiac or cellular level. We chose “2 weeks” because we wanted to evaluate the cellular response at a very initial hypertrophic phase before LV systolic function is worsened. And we found that LV relaxation disturbance already occurred in association with Ca^{2+} leak before onset of LV systolic dysfunction in pressure-overload induced hypertrophy. We are not sure that “2 weeks TAC” is the most suitable for evaluating the initial phase hypertrophy.

2. Fig 2C F/Fo for baseline and 2wk TAC.. Is there shift in time to activation? or is this simply an alignment issue. The upstroke of the transient is offset in the mutant. Clarification of this difference would improve the interpretation of this mutant.

We are sorry that Ca^{2+} transient graph was slightly shifted just for clarification of the upstroke. We corrected the figure without shift.

3. For all the imaging-based experiments, I was surprised that there were no negative controls or images for any of the studies. Given that the data are presented as colocalization between molecules (Cam/RyR; or cyto/nuc) negative controls with their data are needed. This would be for fig3d, 5b, 5d, 6 and 7.

We obtained negative controls for Figure 3d, 5b, 5d, 6 and 7, and put them in Supplementary Fig. 2 and new Fig. 4D, 6B, 6D, 7, 8.

4, the RNAseq data brings a novel approach to all of the biochemistry and imaging studies. however as presented it is not very well described or insightful. For instance, why not provide targeted transcriptomic analysis for all myosins, gata's, NFAT, nkx members, RyR1-3, CAM, CAMKII, etc. As presented a network confirming ANF is up, is not worth an entire figure panel.

We inserted actin-myosins pathway that was detected by IPA pathway analysis into Fig.5D, and moved the signaling pathway of ANF and Calcineurin to supplementary Fig.5A. NFAT pathway detected as a third network was added into supplementary Fig.5B. GATA4, NKX2-5 and MEF2C pathway was suggested by upstream pathway analysis (supplementary Fig.6). The gene expressions of RyR1-3 and CAM were presented in supplementary Fig.3. The pathway of CAMKII was undetected in analysis based on the data of RNA-seq

We inserted the supplementary Fig.3-6, and the sentences were added into Results section.

In the network analysis, activation of cardiomyocytes contraction by actin-myosin complex was suggested as a top score (Fig. 5D), and the activation was indicated to be caused by intracellular Ca²⁺ released by RyR2 from data of KEGG and Gene ontology analyses (supplementary Fig. 4A-C).

Among RyR superfamily, RYR2 was mainly expressed in mice hearts, and the expressions were almost constant in the treated mice (supplementary Fig.3). The expression of *Calml1*, which encodes CaM was enhanced with TAC in WT mice hearts, and expression was disappeared those in V3599K mice (supplementary Fig.3).

The signaling pathway of *Nfat* (nuclear factor of activated T cells, NFAT) family indicated as a third network was associated with ERK1 activation involving in hypertrophic change (supplementary Fig.5B). The upstream analysis using the gene group indicating the Ca²⁺-dependent hypertrophic pathways, detected a transcription factor myocyte enhancer factor 2 (MEF2), and the signaling pathway was suggested to participate in the induction of the hypertrophic genes such as *Acta1*, *Myh7*, and *Nppa* through GATA binding protein 4 (GATA4) activation (supplementary Table1 and supplementary Fig.6).

Minor issues:

1. in the title of the first two paragraphs (line 102 and line 118), there is no actual data showing 'Enhanced caM binding to RyR2...'. I believe the authors are inferring from their previous paper, and these titles could be edited to reflect what is actually in figures 1 and 2.

We changed “Enhanced CaM binding to RyR2” to “RyR2 V3599K mutation”.

2. There are only a few grammatical/stylistic errors in the paper, which would likely be picked up by an typesetting team. For instance, line 320, what does 'it' refer to; the inhibitory action of dantrolene?

We corrected “it” to “the inhibitory action of dantrolene against hypertrophy”.

3. Add molecular weights for bands in Fig 3e.

We corrected Fig.3e (new Fig.4E).

4. The last paragraph, for this reviewer, was 'wordy' and did not express the manuscript clearly.

We corrected the last paragraph as follows: “In conclusion, cardiac hypertrophy/heart failure induced by pressure overload is caused by ROS-mediated destabilization of RyR2 (defective inter-domain interaction→CaM dissociation→Ca²⁺ leakage), and by increasing the binding affinity of CaM to RyR2 pharmacologically or genetically, the abnormal pathways leading to hypertrophy are inhibited, thereby

suppressing the progression to heart failure.”

Reviewer #3 (Remarks to the Author):

Thank you very much for valuable comments.

1. It is unclear if the V3599K effect is entirely due to enhanced CaM binding. In Fig. 3E, the affinity appears very similar between WT and V3599K. The evaluation of affinity is based on peptide-binding studies from the groups prior JCI Insight manuscript. I don't see any data that this holds true in vivo. The results are certainly interesting, but are there other structural changes that this mutation might engender that are responsible for these effects?

As pointed out by the reviewer, the mechanism by which V3599K mutation prevents the decrease in the affinity for the binding of CaM to RyR2 in pressure-overloaded hearts still remains to be elucidated. We added the following sentences in “Discussion” section.

The mechanism by which the V3599K mutation prevents the reduced affinity of CaM for binding to RyR2 in pressure-overloaded hearts still remains to be elucidated. Given that the affinity of CaM binding to the V3599K-mutant RyR2 was similar to WT RyR2 unless pressure-overload was imposed (Figure 4E), CaM accessibility to the V3599K-mutated CaM binding site may be allosterically enhanced, only when the defective inter-subunit interaction of RyR2 is produced. Since CaM interacts densely with three domains (helix α -1, helix 2b, helix α -9) in RyR2 (15), *in situ* CaM accessibility to its binding site may not be easily enhanced by the V3599K mutation, though the V3599K mutation in the CaM binding domain peptide (3583–3603) markedly enhanced the binding affinity to CaM (17). In contrast, the defective inter-subunit interaction due to pressure-overload may allosterically change the conformational state of the CaM binding region, facilitating CaM binding to the V3599K mutant (not WT) domain in helix α -1. The fact that the CaM binding region is three-dimensionally close to the “zipping interface” supports this idea (Supplementary Figure 8).

2. Along the same lines, I would suggest that the results be contextualized in the

setting of the substantial structural data from Nieng Yan's group that has become available on RyR2, including the CaM-bound structures (Gong et. al. Nature 2019, 572: 347). It is surprising the authors neither discuss nor cite this recent data. A figure should be included demonstrating where the V3599K mutation is in relation to CaM binding. The zipping/unzipping concept is simple but outdated, and the manuscript should be substantially rewritten placing the results within the context of the newer structural data.

Thank you very much for bringing an important point to our attention. We agree that our zipping/unzipping concept is now outdated. We have extensively rewritten our manuscript based on the structural data.

We created new Supplementary Figure 8 and added the following sentences respectively in “Introduction and Discussion” sections.

“Although the zipping-unzipping hypothesis can largely explain the pathogenic mechanism of Ca²⁺ leakage in CPVT or heart failure, high-resolution structures determined by cryo-electron microscopy (cryo-EM) revealed that the CPVT-causing mutations do not target a single interface, instead affecting multiple small domain-domain interfaces, within and across the hotspots (14, 15). In particular, most disease mutations in the N-terminal region of RyR1 and 2 involve inter-domain interfaces within the N-terminal region, and only a small fraction may interact with the central hot spots (16). Therefore, with the atomic level structures in mind, more rigorous analysis is needed to determine the critical role of inter-domain interactions on RyR2 function in heart failure.”

“Inter-domain cross-talk between zipping interface and CaM binding site was based on the reciprocal nature of CaM binding and DPc10 binding to RyR2. Namely, exposing RyR2 to DPc10 inhibits CaM binding to RyR2 and CaM binding to RyR2 reduces DPc10 accessibility to its site (27). Gong et al. more recently reported that Apo-CaM and Ca²⁺-CaM bind to distinct but overlapping sites in an elongated cleft formed by the helical, handle, and central domains in RyR2 (PDB ID: 6JV2 also see ref 15). They found that the N- and C-lobes of Ca²⁺-CaM interact with the C- (3596-3605) and N termini of helix α -1 (3587-3592), respectively, and N-lobe of Ca²⁺-CaM also interacts with Arg2209 in helix 2b of handle domain and Met3820 in helix α -9. According to their cryo-EM structures of RyR2, the “zipping interface” between N-terminal and central domains appears to be smaller than we have expected, and also it is located at the inter-subunit interface of RyR2, which is close to CaM binding region

(Supplementary Figure 8). Therefore, the “core domain unzipping” between N-terminal (1-220) and central (2250-2500) domains may cause defective inter-subunit interaction, allosterically induce the conformational change at CaM binding region, thereby displacing CaM in failing hearts. Very interestingly, the binding site of dantrolene (601-620) (25) is just adjacent to that of another RyR2 stabilizer, K201 (JTV519), which we first discovered a stabilizing effect on the RyR2 channel (29) and identified the binding site (2114-2149) (30), suggesting that both drugs commonly correct defective CaM-RyR2 interactions in diseased hearts (Supplementary Figure 8).”

3. I find it confusing that the authors repeatedly claim that the V3599K model shows that compensatory hypertrophy is not physiological but pathological. I think this is fairly well established. The citations used by the authors are several decades old. More recent literature (e.g. reviewed in Nat Rev Cardiol. 2018 Jul;15(7):387) show that the adaptive responses under pressure overload have very different genetic signatures compared to exercise or other forms of “physiological” hypertrophy. I am not entirely sure what new point the authors are trying to make here.

Thank you very much again for providing the updated information. We added these sentences in “Discussion” section.

“Recently, accumulated evidence suggests that chronic pressure-overload induces various pathological signals, including mitochondrial dysfunction, metabolic reprogramming, impaired Ca²⁺ handling, altered sarcomere structure, fibrosis, and that the adaptive response under pressure-overload has very different genetic characteristics compared to exercise and other forms of physiological hypertrophy (28). However, it is difficult to determine whether changes in these pathological signals are the primary cause or secondary effect in many forms of hypertrophy. This study clearly demonstrated that aberrant Ca²⁺ release due to CaM displacement from RyR2 is a primary cause of pressure-overload induced hypertrophy, and that prevention of Ca²⁺ leak by enhancing CaM binding to RyR2 completely inhibits hypertrophy. Moreover, based on the findings that V3599K mice showed much better cardiac function and prognosis after TAC than WT mice, it is strongly suggested that cardiac hypertrophy is indeed a maladaptive response from the beginning and should be corrected as early as possible after pressure-overload is applied to cardiac muscle.”

4. The authors see neither physiological nor pathological hypertrophy following TAC (though if you look carefully at Fig. 1B and 2B there may be a very mild, blunted response). If there is not substantial cellular or organ level hypertrophy, how do these hearts generate the same peak pressures? Is it a change in Frank-Starling mechanism? Was TAC not as effective in the transgenic mice? There needs to be some physical mechanism to handle the increased afterload induced by TAC, but this is neither discussed nor presented in the data here. It would be beneficial to include some organ, cellular, or molecular data to suggest how the transgenic hearts adapt to the increased afterload.

We performed additional experiments to assess the effect of V3599K mutation on *in vivo* LV pressure-volume relationship. Since we do not have the apparatus to measure LV volume, we calculated it from the data of LV internal diameter measured by echo-cardiograms. Then, we noticed that “the slope of LV end-systolic pressure-volume relationship (Ees) decreased 8 weeks after TAC in WT mice, indicating decreased LV contractility. In contrast, the Ees became rather steeper 8 weeks after TAC in V3599K mice than in Sham (V3599K) mice, indicating increased LV contractility (Figure 2).”

This increased contractility may explain why the V3599K heart can adapt to chronic pressure-overload without stretching. However, it is unclear why LV contractility increases in the V3599K hearts. The signaling pathways antagonizing pathological hypertrophic responses, generally observed in “physiological hypertrophy”, might be involved in the increase in LV contractility; e.g. cell survival signaling, increased energy production and efficiency, antioxidant systems, mitochondrial quality control, and cardiomyocyte proliferation and regeneration (28). In particular, given that there was no cellular hypertrophy, and there was no difference in sarcomeric shortening before and after TAC in V3599K cardiomyocytes, cell proliferation processes might be involved in the increase in LV contractility of V3599K hearts.

We added these sentences in “Results and Discussion” sections.

5. The data in Fig. 5 on direct pressure load on cardiomyocytes is intriguing but hard to interpret. I couldn't find a detailed description, citation, or validation of the “air compression” system. I'm not sure how this is supposed to mimic pressure overload and though the results are consistent with their hypothesis, the entire experiment is confusing and I couldn't really place any confidence in it.

We explained the detailed method for air compression on cells in “Methods” section and new Supplementary Figure 7. Although air compression to isolated cells is not the same pressure-overload as that to myocardium by TAC, this method enables us to evaluate the effect of just pressure-overload (without stretch) on cellular function and Ca²⁺ handling. And, to our knowledge, this is the first study showing that only pressure-overload causes ROS-mediated Ca²⁺ leak through RyR2 that can be rescued by normalizing CaM-RyR2 interaction.

6. The survival data for sham V3599K in Fig. 1E is missing. Along these same lines, I wondered if there was some underlying pathology induced at baseline by the V3599K mutation? The authors might speculate why this mutation has not occurred spontaneously if its effects are largely beneficial?

We added the survival data (100 % survival) for sham V3599K in new Figure 1E. We confirmed there was no apparent structural and functional change at baseline between WT and V3599K mice. We have no idea if this mutation occurred spontaneously. If either mutation or SNP at V3599K occurs in some groups of people, it is interesting to know if they are resistant to hypertrophy or heart failure.

7. Contrast in a lot of the images is poor (e.g. Fig 5B) and scale bars are missing in most.

We corrected the images and added scale bars.

8. The speculation on the usefulness of dantrolene was welcome. It certainly seems like an intriguing option for a potential subset of HF patients.

Thank you very much for your encouraging comments.

Reviewers' Comments:

Reviewer #2:

Remarks to the Author:

The authors addressed my initial concerns in a satisfactory manner in this revised manuscript. I recommend publication at this stage.

Reviewer #3:

Remarks to the Author:

The authors have largely addressed my concerns in the new revision. However, one important question still remains. The methods section does not really include any details of the air compression system -- how cells were prepared, how many were used per assay, etc., while Supplementary Figure 7 only includes data about the equipment and analysis pathway. What remains missing, and still is quite confusing to me, is how the air pressure is actually transduced onto cardiomyocytes. As mentioned previously, there aren't prior publications or detailed methods to describe the validity of this methodology. At the very least, the authors should include a detailed, descriptive cartoon or picture of the recording chamber showing how cells are placed, where the air pressure is coming from, and how it is kept from simply blowing off the media and cells from within the chamber. The methods section should describe in more detail how the experiment is performed: e.g How big is the chamber relative to the cells? Is the chamber sealed? How many cells are used? How much media covers the cells? How is the pressure kept uniform across the cells? Do the cells move during periods of increased pressure? I couldn't find anything online about the Labtec SKPTC-35 chamber the authors mention. Although I appreciate the description provided, I still don't see how controlled air pressure is delivered to cardiomyocytes without simply causing the media and cells to be displaced.

Otherwise, I have no remaining concerns about the manuscript.

Dear Editor,

We very much appreciate valuable comments provided by you and the reviewers. In response to comments of the reviewer #3, we have made additional revisions. Below are point-by-point descriptions of the revisions we have made, which, I hope, make this manuscript suitable for publication. For the reviewers' convenience, the revised portions are marked in red in the text.

Responses to the reviewer #3

Thank you very much for valuable comments.

The methods section does not really include any details of the air compression system -- how cells were prepared, how many were used per assay, etc., while Supplementary Figure 7 only includes data about the equipment and analysis pathway. What remains missing, and still is quite confusing to me, is how the air pressure is actually transduced onto cardiomyocytes. As mentioned previously, there aren't prior publications or detailed methods to describe the validity of this methodology. At the very least, the authors should include a detailed, descriptive cartoon or picture of the recording chamber showing how cells are placed, where the air pressure is coming from, and how it is kept from simply blowing off the media and cells from within the chamber. The methods section should describe in more detail how the experiment is performed: e.g How big is the chamber relative to the cells? Is the chamber sealed? How many cells are used? How much media covers the cells? How is the pressure kept uniform across the cells? Do the cells move during periods of increased pressure? I couldn't find anything online about the Labtec SKPTC-35 chamber the authors mention. Although I appreciate the description provided, I still don't see how controlled air pressure is delivered to cardiomyocytes without simply causing the media and cells to be displaced.

We revised the scheme and added detailed description of the acute pressure overload system in Supplementary figure 7, and also "Methods section" in the text.

Supplementary Figure 7. Acute pressure-overload system by air compression applied to isolated cardiomyocytes.

Monitor shows representative pressure waves due to systolic pressure load (upper) and diastolic pressure load (lower) triggered by electrical stimulation (1Hz).

The mixed gas (5% O₂, 95% CO₂) compression system on cardiomyocytes is composed of 1) tank made of stainless steel for generating air pressure (SKBT-5L, LABTEC, Fukuoka,

Japan) **2)** pressure load timer device (SKPTC-H5CZ, LABTEC, Fukuoka, Japan) **3)** Stainless steel chamber for direct pressure load on cultured cardiomyocytes (SKPTC-35, LABTEC, Fukuoka, Japan) **4)** Mikro-Tip Catherer Pressure Transducers (Millar Instruments, Inc, Houston, USA) **5)** Blood Pressure Amplifiers (FE117 AD Instruments, Colorado Springs, USA) **6)** PowerLab 8/35 (AD Instruments, Colorado Springs, USA) **7)** Personal computer with LabChart physiological data analysis software (AD Instruments, Colorado Springs, USA) **8)** MyoPacer Field Stimulator (ION Optix, Westwood, USA)._

The tank connected to the high pressure gas cylinder (5% O₂, 95%CO₂) can freely control magnitude of the pressure load on the cardiomyocytes in the chamber, and the pressure load timer device can freely control the duration and timing of the pressure load by opening or closing the valve in the apparatus, which is triggered by electrical stimulation from MyoPacer Field Stimulator. The controlled air pressure is delivered into single entrance of the pressure load chamber.

As shown in the lower panel, 100 to 200 isolated cardiomyocytes attached to the laminin-coated 35mm-diameter glass culture dish are incubated in 2000 µl of Tyrode's solution in the presence 2, 3-butanedione monoxime (BDM) to eliminate the effect of contraction on hypertrophic signaling. The custom pressure load chamber consists of a stainless steel top and bottom lid. The outer diameter of the chamber is 75 mm and the inner diameter is 35 mm. The 35 mm-diameter dish is exactly put in the lower lid and the top lid is firmly fixed. The cardiomyocytes in the glass culture dish are sealed in a stainless steel chamber, so the pressure throughout the cardiomyocytes is kept uniform. There is no cardiomyocyte movement during the acute pressure load on the cardiomyocytes. Mikro-Tip Catherer Pressure Transducers (Millar Instruments, Inc, Houston, USA) is set in the sealed space made of the pressure load chamber. Therefore, the magnitude of the pressure load on the cardiomyocytes in the chamber is accurately measured by an established catheter-tip manometer method, and monitored by a personal computer via Blood Pressure Amplifiers PE117 and PowerLab 8/35. Pressure-overload (150 mmHg, 250 ms) was applied to cardiomyocytes with 1 Hz electrical pacing during the (systolic) Ca²⁺ transient phase or (diastolic) static phase. Ca²⁺ sparks were measured after 10 minutes of pressure overload. The acquisition and analysis of the data were conducted by LabChart (physiological data analysis software).

In “Methods” section in page 15, we added the following sentences; “Supplementary Fig. 7 shows the scheme and detailed description of the acute pressure overload system. Pressure-overload (150 mmHg, 250 ms) was applied to cardiomyocytes with 1 Hz electrical pacing during the (systolic) Ca²⁺ transient phase or (diastolic) static phase. Ca²⁺ sparks were measured after 10 minutes of pressure overload. Endogenous

CaM-RyR2 binding experiments and hypertrophic signaling assay (HDAC, NFAT) were performed after 30 minutes of pressure overload. Data were recorded at each assay using 10 to 20 cardiomyocytes per dish.”